# Unifying Self-Supervised Clustering and Energy-Based Models

**Emanuele Sansone**                                          *emanuele.sansone@kuleuven.be*
*Department of Electrical Engineering (ESAT)*
*KU Leuven*

**Robin Manhaeve**                                          *robin.manhaeve@kuleuven.be*
*Department of Computer Science*
*KU Leuven*

**Reviewed on OpenReview:** *https://openreview.net/forum?id=NWOuKe6IZa*

## Abstract

Self-supervised learning excels at learning representations from large amounts of data. At the same time, generative models offer the complementary property of learning information about the underlying data generation process. In this study, we aim at establishing a principled connection between these two paradigms and highlight the benefits of their complementarity. In particular, we perform an analysis of self-supervised learning objectives, elucidating the underlying probabilistic graphical models and presenting a standardized methodology for their derivation from first principles. The analysis suggests a natural means of integrating self-supervised learning with likelihood-based generative models. We instantiate this concept within the realm of cluster-based self-supervised learning and energy models, introducing a lower bound proven to reliably penalize the most important failure modes and unlocking full unification. Our theoretical findings are substantiated through experiments on synthetic and real-world data, including SVHN, CIFAR10, and CIFAR100, demonstrating that our objective function allows to jointly train a backbone network in a discriminative and generative fashion, consequently outperforming existing self-supervised learning strategies in terms of clustering, generation and out-of-distribution detection performance by a wide margin. We also demonstrate that the solution can be integrated into a neuro-symbolic framework to tackle a simple yet non-trivial instantiation of the symbol grounding problem. The code is publicly available at `https://github.com/emsansone/GEDI.git`.

## 1 Introduction

Self-supervised learning (SSL) has achieved remarkable results in recent years thanks to its ability to learn high-quality representations from large amounts of unlabeled data (Balestriero et al., 2023). At the same time, generative models have provided valuable insights into the unknown generative processes underlying data. The synergy between these two distinct areas of machine learning holds great potential, for example by leveraging knowledge of the underlying generative process to learn more robust representations, or by using learned representations to synthesize new data. However, a principled theory and methodology that bridges self-supervised learning and generative modeling is missing. In this work, we take a significant step toward closing this gap by demonstrating, for the first time, the feasibility of learning a self-supervised clustering model in a generative manner. To achieve this, we address three key challenges: **formulation**, **integration**, and **unification**.

The first challenge arises from the diverse range of SSL objectives proposed in recent years and the absence of a common perspective. We tackle this by providing a probabilistic interpretation of these objectives through their underlying probabilistic graphical models. This perspective reveals a common methodology

for deriving existing objectives from first principles. The resulting probabilistic framework naturally paves the way to connect SSL with likelihood-based generative models, thereby addressing the second challenge of integration.

The final challenge, unification, involves reconciling the variety of architectural heuristics commonly used to prevent trivial representations - heuristics that often hamper generative capabilities. We overcome this by introducing a novel objective, the **GEnerative DIscriminative lower bound** (GEDI). We prove that GEDI is guaranteed to avoid the triad of failure modes (Sansone, 2023), including representation collapse, cluster collapse, and the problem of label inconsistency with data augmentations. These guarantees eliminate the need for common architectural heuristics and enable training a neural network in both a generative (akin to energy-based models) and a discriminative manner (similar to SSL clustering). The proposed unified view ultimately leads to improved performance in terms of clustering, generation, and out-of-distribution detection compared to standard cluster-based SSL methods.

We substantiate our theoretical findings through experiments conducted on both toy and real-world datasets. Specifically, our results demonstrate that GEDI can achieve a significant improvement in terms of clustering performance compared to state-of-the-art baselines on SVHN, CIFAR-10 and CIFAR-100. Additionally, in the context of generation performance, GEDI can effectively compete with existing energy-based solutions, whereas traditional SSL approaches fall short. Most importantly, we highlight that the generative nature of GEDI is a crucial aspect that enhances the model's ability to detect out-of-distribution data, surpassing purely discriminative baselines in this regard. Finally, we show that GEDI can be easily integrated into a neuro-symbolic framework like DeepProbLog (Manhaeve et al., 2018) and leverage its clustering nature to learn higher quality symbolic representations when performing symbol grounding (Harnad, 1990; Barsalou, 1999; Manhaeve et al., 2018; 2021; Sansone & Manhaeve, 2022; 2023a;b; Marconato et al., 2023).

The article is structured as follows: In §2, we provide a probabilistic interpretation of three classes of SSL approaches, namely contrastive, negative-free and cluster-based methods. In §3, we showcase the integration of SSL with likelihood-based generative models. In §4, we take a step further and provide an instantiation of the general framework, thus providing the first principled objective to unify cluster-based and energy-based models and that is guaranteed to avoid the triad of failure modes. In §5, we review related work on SSL and in §6 we discuss the experimental analysis. Finally, in §7, we conclude by discussing future research directions for SSL.

## 2 A Probabilistic Formulation of SSL Objectives

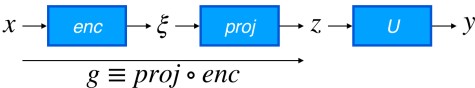

Figure 1: Different function blocks and variables used throughout the analysis of existing SSL approaches.

Let us introduce the random quantities used throughout this section to analyze self-supervised learning approaches. We use $x \in \Omega$, where $\Omega$ is a compact subset of $\mathbb{R}^d$, to identify observed data drawn independently from an unknown distribution $p(x)$ and $x' \in \Omega$ to identify observed data drawn independently from a stochastic augmentation strategy $\mathcal{T}(x'|x)$. We use $\xi \in \mathbb{R}^h$, and $z \in \mathbb{R}^l$, to identify latent representations in a deep network architecture, see Fig. 1) for a visualization. Importantly, the first latent representation is obtained through an encoding function $enc : \Omega \to \mathbb{R}^h$, whereas the second representation is obtained through the composition of $enc$ and a projector function $proj : \mathbb{R}^h \to \mathbb{R}^l$, namely $g : \Omega \to \mathbb{R}^l$. Finally, we introduce a categorical variable $y$ to identify the final output of the deep network architecture.

We focus our analysis on three main classes of self-supervised learning approaches: 1) contrastive, 2) negative-free and 3) cluster-based methods.[1] For each, we construct the underlying probabilistic graphical model (cf. Figure 2). This allows us to derive their corresponding objective functions from first principles and

---

[1] We focus on purely encoder-based techniques and leave the extension to predictive models (e.g. masked autoencoders) for future work.

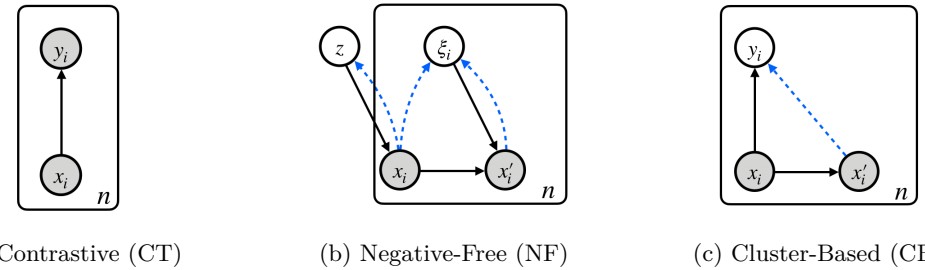

(a) Contrastive (CT)    (b) Negative-Free (NF)    (c) Cluster-Based (CB)

Figure 2: Probabilistic graphical models for the different classes of self-supervised learning approaches. White and grey nodes represent hidden and observed vectors/variables, respectively. Solid arrows define the generative process, whereas blue dashed arrows identify auxiliary posterior densities/distributions. Index $i \in \{1, \ldots, n\}$ is used to identify training samples and their corresponding representations.

highlight that all objectives share a common structure. Further on, we will use these insights to integrate self-supervised learning with generative models.

## 2.1 Contrastive SSL

We demonstrate that contrastive self-supervised learning can be modeled as an instance classification problem, and reveal that the learning objective can be decomposed into two main terms: an entropy and a discriminative term. For the simplicity of the exposition, we focus on one of the main contrastive objectives, InfoNCE (den Oord et al., 2018). Same conclusions can be obtained for other objectives such as CPC (O. Henaff, 2020), SimCLR (Chen et al., 2020), ProtoCPC (Lee, 2022), KSCL (Xu et al., 2022) to name a few, as demonstrated in appendices E to H. In the following, we introduce the ground truth and model predictive distributions and demonstrate how contrastive learning encourages alignment between the two.

**Definition 1** (Constrastive SSL ground truth joint distribution)**.**

$$p(x_{1:n}, y_{1:n}) \equiv \prod_{j=1}^{n} p(x_j)\delta(y_j - j)$$

with $y_j \in \{1, \ldots, n\}$ and $\delta$ a delta function.

**Definition 2** (Constrastive SSL model distribution)**.**

$$p(x_{1:n}, y_{1:n}; \Theta) \equiv \prod_{i=1}^{n} p(x_i)p(y_i|x_i; \Theta)$$

$$p(y_i|x_i; \Theta) \equiv \frac{e^{sim(g(x_i;\theta),g(x_{y_i};\theta))/\tau}}{\sum_{k=1}^{n} e^{sim(g(x_i;\theta),g(x_k;\theta))/\tau}}$$

with $sim : \mathbb{R}^l \times \mathbb{R}^l \to \mathbb{R}$ a similarity function, $\tau > 0$ a temperature parameter and $\Theta = \{\theta, \{x_i\}_i^n\}$ a set of parameters.

The ground truth distribution defines an underlying instance classifier, where each input $x_i$ is associated to a unique natural number identifier through the $\delta$ function. The following Lemma provides an alternative yet equivalent interpretation of contrastive learning (the proof is given in Appendix B).

**Lemma 1.** *Given Definitions 1 and 2, maximizing the InfoNCE objective is equivalent to maximize the following log-likelihood lower bound:*

$$E_{p(x_{1:n}, y_{1:n})}\{\log p(x_{1:n}, y_{1:n})\} \geq \underbrace{-H_p(x_{1:n})}_{Neg.\ entropy\ term} + \underbrace{\mathbb{E}_{\prod_{j=1}^{n} p(x_j)}\left\{ \sum_{i=1}^{n} \log \frac{e^{sim(g(x_i;\theta),g(x_i;\theta))/\tau}}{\sum_{k=1}^{n} e^{sim(g(x_i;\theta),g(x_k;\theta))/\tau}} \right\}}_{discriminative\ SSL\ term\ \mathcal{L}_{CT}(\Theta)}$$

*Moreover, the maximization of this lower bound is equivalent to solve an instance classification problem.*

From the above Lemma, we observe that the contrastive learning objective involves a discriminative and a negative entropy term. Only the discriminative SSL term is usually optimized, due to the fact that the entropy terms does not depend on $\Theta$.

## 2.2 Negative-Free SSL

We demonstrate that negative-free SSL enforces two important properties, namely the decorrelation of features in $z$ and the invariance w.r.t. data augmentation over $\xi$. Furthermore, we reveal that negative-free SSL can be decomposed into two main parts, similarly to contrastive SSL. We focus the analysis on a recent negative-free criterion, namely CorInfoMax (Ozsoy et al., 2022). Similar conclusions can be derived for other negative-free approaches, including Barlow Twins (Zbontar et al., 2021), VicReg (Bardes et al., 2022a;b) and W-MSE (Ermolov et al., 2021) (please refer to Appendix I for further details). In the following, we introduce the ground truth and model distributions and demonstrate how negative-free SSL encourages alignment of their covariance matrices. In effect, this drives the sample covariance matrix toward a diagonal form, thereby decorrelating the features.

**Definition 3** (Negative-free SSL ground-truth distribution)**.**

$$p(z, \xi_{1:n}, x_{1:n}, x'_{1:n}) \equiv p(z) \prod_{i=1}^{n} p(x_i|z) p(\xi_i) p(x'_i|x_i, \xi_i)$$

*with Gaussian priors $p(z) = \mathcal{N}(z|0, I)$, $p(\xi_i) = \mathcal{N}(\xi_i|0, \gamma^{-1}I)$, and assume the following conditional independencies $p(x'_i|x_i, \xi_i) = \mathcal{T}(x'_i|x_i)$ and $p(x_i|z) = p(x_i)$.*

**Definition 4** (Negative-free SSL auxiliary model distributions)**.**

$$q(\xi_i|x_i, x'_i; \Theta) \equiv \mathcal{N}(\xi_i|enc(x_i; \theta) - enc(x'_i; \theta), I)$$
$$q(z|x_{1:n}; \Theta) \equiv \mathcal{N}(z|0, \Sigma)$$

*with $\Sigma \equiv \sum_{i=1}^{n} (g(x_i; \theta) - \bar{g})(g(x_i; \theta) - \bar{g})^T + \beta I$, $\beta > 0$ chosen to ensure the positive definiteness of $\Sigma$ and $\bar{g} = 1/n \sum_{i=1}^{n} g(x_i; \theta)$ and $\Theta = \{\theta\}$.*

We can state the following Lemma with proof provided in Appendix C.

**Lemma 2.** *Given Definitions 3 and 4, maximizing the CorInfoMax objective (Ozsoy et al., 2022) is equivalent to maximize the following log-likelihood lower bound:*

$$E_{p(x_{1:n}, x'_{1:n})}\{\log p(x_{1:n}, x'_{1:n})\} \geq \underbrace{-H_p(x_{1:n})}_{\text{Neg. entropy term}} \underbrace{-E_{p(x_{1:n})}\{KL(q(z|x_{1:n}; \Theta)\|p(z))\}}_{\text{discriminative SSL term } \mathcal{L}_{NF}(\Theta)}$$

$$\underbrace{-\sum_{i=1}^{n} E_{p(x_i)\mathcal{T}(x'_i|x_i)}\{KL(q(\xi_i|x_i, x'_i; \Theta)\|p(\xi_i))\}}_{\text{Continuation of } \mathcal{L}_{NF}(\Theta)} + const$$

*where the first Kullback-Leibler term $KL(q(z|x_{1:n}; \Theta)\|p(z)) \equiv KL(\mathcal{N}(z|0, \Sigma)\|\mathcal{N}(z|0, I))$, the second term $KL(q(\xi_i|x_i, x'_i; \Theta)\|p(\xi_i)) \equiv KL(\mathcal{N}(\xi_i|enc(x_i; \theta) - enc(x'_i; \theta), I)\|\mathcal{N}(\xi_i|0, \gamma^{-1}I))$ and const being a constant for the optimization over $\Theta$.*

In other words, the discriminative terms promote two properties. Indeed, the first term standardizes the sample covariance of the latent representation, thus decorrelating its feature elements, and the the second term promote the invariance to data augmentations. These two properties are common among other negative-free methods, as shown in Appendix I. Notably, the negative-entropy term in the log-likelihood lower bound is not influenced by the optimization.

### 2.3 Cluster-Based SSL

We demonstrate that cluster-based SSL admits a probabilistic interpretation based on the graphical model in Fig. 2(c), thus revealing that the objective can be decomposed in a negative entropy and a discriminative term. The analysis focusses on a recent approach SwAV (Caron et al., 2020), but is more generally applicable to other cluster-based approaches, including DeepCluster (Caron et al., 2018) and SeLA (Asano et al., 2020) to name a few. In the following, we introduce the ground truth and model predictive distributions and demonstrate how contrastive learning encourages alignment between the two.

**Definition 5** (Cluster-Based SSL ground-truth joint distribution)**.**

$$p(x_{1:n}, x'_{1:n}, y_{1:n}) \equiv \prod_{i=1}^{n} p(x_i)\mathcal{T}(x'_i|x_i)p(y_i|x_i; \Theta)$$

with $y_i \in \{1, \ldots, c\}$ being a categorical variable to identify one of c clusters, $p(y_i|x_i; \Theta) = \frac{e^{U_{:y_i}^T G_{:i}/\tau}}{\sum_y e^{U_{:y}^T G_{:i}/\tau}}$, where $U \in \mathbb{R}^{h \times c}$ is the matrix[2] of cluster centers, $G = [g(x_1; \theta), \ldots, g(x_n; \theta)] \in \mathbb{R}^{h \times n}$ is a representation matrix and $\Theta = \{\theta, U\}$ is the set of parameters.

**Definition 6** (Cluster-Based SSL auxiliary clustering distribution)**.**

$$q(y_{1:n}|x'_{1:n}) \equiv \prod_{i=1}^{n} q(y_i|x'_i)$$

We can state the following Lemma (proof is provided in Appendix D)

**Lemma 3.** *Given Definitions 5 and 6, maximizing the SwAV objective (Caron et al., 2020) is equivalent to maximize the following log-likelihood lower bound:*

$$E_{p(x_{1:n}, x'_{1:n})}\{\log p(x_{1:n}, x'_{1:n})\} \geq \underbrace{-H_p(x_{1:n})}_{Neg.\ entropy\ term} + const$$
$$+ \underbrace{\sum_{i=1}^{n} \mathbb{E}_{p(x_i)\mathcal{T}(x'_i|x_i)}\left\{\mathbb{E}_{q(y_i|x'_i)}\log p(y_i|x_i; \Theta) + H_q(y_i|x'_i)\right\}}_{discriminative\ SSL\ term\ \mathcal{L}_{CB}(\Theta)}$$

An important aspect of cluster-based SSL is that the categorical variables $y_{1:n}$ are unobserved. A naive maximization of $\mathcal{L}_{CB}(\Theta)$ can lead to obtain trivial solutions like the one corresponding to uninformative predictions, namely $q(y_i|x'_i) = p_\gamma(y_i|x_i) = \text{Uniform}(\{1, \ldots, c\})$ for all $i = 1, \ldots, n$. Fortunately, the problem can be avoided and solved exactly using an alternating optimization strategy based on the Sinkhorn-Knopp algorithm, where the alternation occurs between the optimization over the auxiliary distribution and $\Theta$. This is indeed the procedure used in several cluster-based SSL approaches, like SwAV (Caron et al., 2020), DeepCluster (Caron et al., 2018) and SeLA (Asano et al., 2020) to name a few. We will see later on that we can obtain a different lower bound for cluster-based SSL which doesn't require asymmetries, in the form of alternating optimization, stopping gradients or momentum encoders.

## 3 Integrating SSL and Likelihood-Based Generative Models

In all three classes of SSL approaches (cf. Lemmas 3, 6 and 9), the expected data log-likelihood can be lower bounded by the sum of two contribution terms, namely a negative entropy $-H_p(x_{1:n})$ and a discriminative log-likelihood term, chosen from $\mathcal{L}_{CT}(\Theta), \mathcal{L}_{CB}(\Theta)$ and $\mathcal{L}_{NF}(\Theta)$. A connection to generative models emerges by additionally lower bounding the negative entropy term, namely:

$$-H_p(x_{1:n}) = \mathbb{E}_{p(x_{1:n})}\{\log p(x_{1:n})\}$$

---

[2]We use subscripts to select rows and columns. For instance, $U_{:y}$ identify $y-$th column of matrix $U$.

$$= \sum_{i=1}^{n} \mathbb{E}_{p(x_i)}\{\log p(x_i)\}$$

$$= \sum_{i=1}^{n} \left[ \mathbb{E}_{p(x_i)}\{\log p_\Psi(x_i)\} + KL(p(x_i)\|p_\Psi(x_i)) \right]$$

$$\geq \underbrace{\sum_{i=1}^{n} \mathbb{E}_{p(x_i)}\{\log p_\Psi(x_i)\}}_{-CE(p, p_\Psi)} \tag{1}$$

where $p_\Psi(x)$ is a generative model parameterized by $\Psi$. Notably, the relation in (1) can be substituted in any of the objectives previously derived for the different SSL classes, thus allowing to integrate generative and SSL models into a single Bayesian framework. It is important to mention that much can be gained by synergistically optimizing the resulting hybrid objectives. For instance, the recent work EBCLR (Kim & Ye, 2022) proposes a specific instantiation of this general idea, by combining energy-based models with contrastive SSL approaches and showcasing the improved discriminative performance. EBCLR can be obtained by observing that the conditional density in Definition 1 can be decomposed into joint and marginal densities (similarly to what is done in (Grathwohl et al., 2020) for a supervised classifier):[3]

$$p(y|x;\Theta) \equiv \frac{e^{sim(g(x),g(x_y))/\tau}}{\sum_{k=1}^{n} e^{sim(g(x),g(x_k))/\tau}} \Rightarrow \begin{cases} p(y,x;\Theta) = \frac{e^{sim(g(x),g(x_y))/\tau}}{\Gamma(\Theta)} \\ p(x;\Theta) = \frac{\sum_{k=1}^{n} e^{sim(g(x),g(x_k))/\tau}}{\Gamma(\Theta)} = \frac{e^{-E(x;\Theta)}}{\Gamma(\Theta)} \end{cases} \tag{2}$$

where $E(x,\Theta) = -\log \sum_{j=1}^{n} e^{sim(g(x_{\ell_j}),g(x))/\tau}$ defines the energy score of the marginal density. By choosing $p_\Psi(x) = p(x;\Theta)$ and $sim(z,z') = -\|z - z'\|^2$ in Eq. (2), one recovers the exact formulation of EBCLR (Kim & Ye, 2022). Moreover, this integrated framework is not bound to the specific family of energy-based models, rather one could in principle specify any likelihood-based model (e.g. VAE, diffusion model, normalizing flow) in Eq. (1), consequently giving rise to a wide spectrum of possible hybrid solutions. Indeed, some recent works have started to propose solution aiming towards the joint integration of VAEs and contrastive SSL (Gatopoulos & Tomczak, 2020; Zhue et al., 2020; Wu et al., 2023; Nakamura et al., 2023). Therefore, our work generalizes these specific views to a broader class of SSL approaches and likelihood based generative models. In the subsequent sections, we are going to demonstrate an instantiation of this general view based on SSL clustering and energy-based models. Indeed, we are going to provide a unified theory allowing to learn a backbone classifier network to jointly exploit its generative and discriminative properties.

## 4 Unifying Self-Supervised Clustering and Energy-Based Models

Before demonstrating the unification, we provide an alternative lower bound for cluster-based SSL to the one obtained in Lemma 9, enabling to simplify the architectural design of the neural network, while ensuring the avoidance of the main failure modes. Then, we provide our new GEnerative and DIscriminative (GEDI) Lower Bound.

### 4.1 Lower Bound for Cluster-Based SSL

We state the following proposition and leave the proof to Section F in the Supplementary Material.

**Definition 7.** *Consider the same conditions in Definition 5, except for choosing identical auxiliary and predictive distributions, namely $q(y|x) \equiv p(y|x;\Theta) \equiv Softmax(U^T g(x)/\tau)$.*

Note that the choice of identical auxiliary and predictive distributions is a key difference with respect to existing cluster-based SSL. For instance, SwAV avoids learning trivial solutions when optimizing the lower bound in Lemma 9 by introducing asymmetric distributions. Specifically, it considers $p(y|x;\Theta) \equiv$ Softmax$(U^T g(x)/\tau)$ and $q(y|x') \equiv$ Sinkhorn(StopGrad$(U^T g(x')/\tau)$), where Sinkhorn and StopGrad are two

---

[3]For the sake of notation simplicity, we omit the dependence on $\theta$.

operators performing the Sinkhorn-Knopp algorithm and stopping the gradients, respectively. Therefore, our choice constitutes a simplification of the architectural design, but this relaxation requires the development of a new objective to guaranteeing the avoidance of the main training failure modes. The following proposition provide an alternative lower bound for cluster-based SSL (the proof is given in Appendix J, whereas the analysis about failure modes is deferred to Section 4.3)

**Proposition 1.** *Given Definition 7, the expected data log-likelihood for the probabilistic graphical model in Fig. 2(c) can be alternatively lower bounded as follows:*

$$E_{p(x_{1:n},x'_{1:n})}\{\log p(x_{1:n},x'_{1:n})\} \geq -H_p(x_{1:n}) - \underbrace{\sum_{i=1}^{n} \mathbb{E}_{p(x_i)\mathcal{T}(x'_i|x_i)}\left\{CE(p(y_i|x'_i;\Theta),p(y_i|x_i;\Theta))\right\}}_{\mathcal{L}_{INV}(\Theta)}$$

$$\underbrace{-\sum_{i=1}^{n} CE\left(p(y_i),\frac{1}{n}\sum_{j=1}^{n}p(y_j=y_i|x_j;\Theta)\right)}_{\mathcal{L}_{PRIOR}(\Theta)} + const \tag{3}$$

*Additionally, the corresponding maximum value for the last two addends in Eq. (3) is given by the following inequality:*[4]

$$\mathcal{L}_{INV}(\Theta) + \mathcal{L}_{PRIOR}(\Theta) \leq -H_p(y_{1:n}) \tag{4}$$

The above proposition has interesting implications. First of all, by maximizing the discriminative term $\mathcal{L}_{INV}(\Theta)$ with respect to $\Theta$, we enforce two properties, namely: (i) label invariance, as we ensure that the predictive distributions of the discriminative model for a sample and its augmented version match each other and (ii) confident predictions, as maximizing the cross-entropy forces also to decrease the entropy of these distributions.[5] Secondly, by choosing a uniform prior, viz. $p(y_i) = \text{Uniform}(\{1,\ldots,c\})$, and by maximizing $\mathcal{L}_{PRIOR}(\Theta)$ with respect to $\Theta$, we ensure to obtain a balanced cluster assignment. This is also commonly done by approaches based on optimal transport objectives and corresponding surrogates, typically emmployed in cluster-based SSL (Cuturi, 2013; Caron et al., 2018; 2020; Amrani et al., 2022).

### 4.2 GEDI Lower Bound

For the GEDI instantiation, we derive the lower bound on the expected log-likelihood by exploiting the bound in Eq. (1) and the one in Proposition 1:

$$\mathbb{E}_{p(x_{1:n})}\{\log p(x_{1:n};\Theta)\} \geq \underbrace{\mathcal{L}_{GEN}(\Psi)}_{\text{Generative term } -CE(p,p_\Psi)} + \underbrace{\mathcal{L}_{INV}(\Theta) + \mathcal{L}_{PRIOR}(\Theta)}_{\text{Discriminative terms}} \tag{5}$$

additionally we decompose the discriminative model $p(y|x;\Theta)$ to obtain $p_\Psi$ in a similar manner to what we have already done in Eq. (2), namely:

$$p(y,x;\Theta) = \frac{e^{U_{:y}^T g(x)/\tau}}{\Gamma(\Theta)}$$

$$p(x;\Psi) \underbrace{=}_{\Psi=\Theta} p(x;\Theta) = \frac{\sum_{y=1}^{c} e^{U_{:y}^T g(x)/\tau}}{\Gamma(\Theta)} = \frac{e^{-E(x;\Theta)}}{\Gamma(\Theta)} \tag{6}$$

where $E(x;\Theta) = -\log\sum_{y=1}^{c} e^{U_{:y}^T g(x)/\tau}$. We will shortly analyze the properties of the different objective terms in Eq. (5). For the moment, we finalize the GEDI instantiation by devising the corresponding training algorithm.

---

[4]Here, we assume that the predictive model $p(y|x;\Theta)$ has enough capacity to achieve the optimal solution.

[5]Indeed, recall that $CE(p,q) = H_p + KL(p\|q)$. Therefore, maximizing $-CE(p,q)$ forces to have both $KL(p\|q) = 0$ and $H_p = 0$.

---

**Algorithm 1:** GEDI Training.

---

**Input:** $x_{1:n}$, $x'_{1:n}$, Iters, SGLD and Adam optimizer hyperparameters;
**Output:** Trained model $\Theta$;
**For** iter $= 1, \dots,$ Iters;
    Generate samples from $p_\Theta$ using SGLD;
    Estimate $\Delta_1 \Theta = \nabla_\Theta \mathcal{L}_{GEN}(\Theta)$ using Eq. (7);
    Compute $\Delta_2 \Theta = \nabla_\Theta \mathcal{L}_{INV}(\Theta)$;
    Compute $\Delta_3 \Theta = \nabla_\Theta \mathcal{L}_{PRIOR}(\Theta)$ ;
    $\Delta\Theta \leftarrow \sum_{i=1}^{3} \Delta_i \Theta$;
    $\Theta \leftarrow$ Adam maximizing using $\Delta\Theta$;
**Return** $\Theta$;

---

**Learning a GEDI model.** We can train the GEDI model by jointly maximizing the objective in Eq. (5) with respect to the parameters $\Theta$ through gradient-based strategies. The overall gradient includes the summation of three terms, viz. $-\nabla_\Theta CE(p, p_\Theta)$, $\nabla_\Theta \mathcal{L}_{INV}(\Theta)$ and $\nabla_\Theta \mathcal{L}_{PRIOR}(\Theta)$. While the last two gradient terms can be computed easily by leveraging automatic differentiation, the first one must be computed by exploiting the following identities (obtained by simply substituting Eq. (6) into the definition of cross-entropy and expanding $\nabla_\Theta \Gamma(\Theta)$):

$$-\nabla_\Theta CE(p, p_\Theta) = \sum_{i=1}^{n} \mathbb{E}_{p(x_i)} \left\{ \nabla_\Theta \log \sum_{y=1}^{c} e^{U_{:y}^T g(x_i)/\tau} \right\} - n \nabla_\Theta \log \Gamma(\Theta)$$

$$= \sum_{i=1}^{n} \mathbb{E}_{p(x_i)} \left\{ \nabla_\Theta \log \sum_{y=1}^{c} e^{U_{:y}^T g(x_i)/\tau} \right\} - n \mathbb{E}_{p_\Theta(x)} \left\{ \nabla_\Theta \log \sum_{y=1}^{c} e^{U_{:y}^T g(x)/\tau} \right\} \quad (7)$$

Importantly, the first and the second expectations in Eq. (7) are estimated using the training and the generated data, respectively. To generate data from $p_\Theta$, we use a sampler based on Stochastic Gradient Langevin Dynamics (SGLD), thus following recent best practices to train energy-based models (Xie et al., 2016; Nijkamp et al., 2019; Du & Mordatch, 2019; Nijkamp et al., 2020). The whole learning procedure is summarized in Algorithm 1.

**Computational requirements.** When comparing our GEDI instantiation with traditional SSL training, more specifically to SwAV, we observe two main differences in terms of computation. Firstly, our learning algorithm does not require to run the Sinkhorn-Knopp algorithm, thus saving computation. Secondly, our GEDI instantiation requires additional forward and backward passes to draw samples from the energy-based model $p_\Theta$. However, the number of additional passes through the discriminative model can be limited by the number of SGLD iterations, necessary to generate data (cf. Experiments).

### 4.3 Analysis of Loss Landscape and the Triad of Failure Modes

Asymmetries have been playing an important role in self-supervised learning in order to avoid trivial solutions/failure modes (Wang et al., 2022). Here, we formalize three main failure modes for cluster-based SSL. Then, we study the GEDI loss landscape and show that these undesired trivial solutions are not admitted by our objective. This result holds without introducing asymmetries in the optimization procedure and/or network architecture.

Let's start by defining the most important failure modes, namely:

**Definition 8** (Failure Mode 1 - Representational Collapse). *There exists a constant vector $k \in \mathbb{R}^h$ such that for all $x \in \mathbb{R}^d$, $g(x) = k$.*

**Definition 9** (Failure Mode 2 - Cluster Collapse). *There exists a cluster $j \in \{1, \dots, c\}$ such that for all $x \in \mathbb{R}^d$, $p(y = j|x; \Theta) = 1$.*

**Definition 10** (Failure Mode 3 - Label Inconsistency). *For all possible permutations $\pi : \{1, \dots, c\} \to \{1, \dots, c\}$, a dataset $\mathcal{D} = \{(x_i, t_i, t'_i)\}_{i=1}^{n}$, its permuted version $\mathcal{D}^\pi = \{(x_i, t_{\pi(i)}, t'_i)\}_{i=1}^{n}$ and a loss $\mathcal{L}(\Theta; \cdot)$,*

*evaluated at one of the two datasets, we have that $\mathcal{L}(\Theta; \mathcal{D}) = \mathcal{L}(\Theta; \mathcal{D}^\pi)$. For GEDI, $t_i \doteq U^T g(x_i)$ and $t_i' \doteq U^T g(x_i')$.*

In other words, Definition 8 considers the case where the encoder maps (collapses) every input to the same output. Definition 9 considers the situation where the predictive model assigns all samples to the same cluster with high confidence. And Definition 10 considers the case where a hypothetical adversary swaps the predictions made by the model on different pair of inputs. Ideally, we would like to have an objective that does not admit these failure modes.

Now, we state the properties of the loss landscape of GEDI with the following theorem (we leave the proof to Appendix K):

**Theorem 1.** *Given definitions 8-10, the following statements tells for a particular loss, which failure modes are admissible solutions:*

    *a. $\mathcal{L}_{GEN}(\Theta)$ admits failure modes 2 and 3.*

    *b. $\mathcal{L}_{INV}(\Theta)$ admits failure modes 1 and 2.*

    *c. $\mathcal{L}_{PRIOR}(\Theta)$ admits failure modes 1 and 3.*

*Proof.* (Sketch) The proof proceeds by applying the definitions of the various failure families (Definitions 8–10), evaluating the loss terms in each case, and verifying whether their values correspond to global minima. Intuitively, this is feasible because each loss term defines a notion of distance between the ground truth and model distributions. $\square$

Importantly, Theorem 1 tells us that $\mathcal{L}_{GEN}(\Theta)$ can be used to penalize representational collapse, $\mathcal{L}_{INV}(\Theta)$ can be used to ensure that cluster assignments are consistent with data augmentation, while $\mathcal{L}_{PRIOR}(\Theta)$ can be used to penalize cluster collapse. Consequently, by maximizing the objective in Eq. (5), we are guaranteed to learn solutions which are non-trivial. A table summarizing all these properties is given below.

Table 1: Summary of loss landscape

| Does ↓ penalize →? | Repr. collapse | Clus. collapse | Lab. Inconst. |
|---|:---:|:---:|:---:|
| $\mathcal{L}_{GEN}(\Theta)$ | **Yes** | No | No |
| $\mathcal{L}_{INV}(\Theta)$ | No | No | **Yes** |
| $\mathcal{L}_{PRIOR}(\Theta)$ | No | **Yes** | No |
| Eq. (5) | **Yes** | **Yes** | **Yes** |

## 5 Related Work

We organize the related work according to different objective categories, namely contrastive, cluster-based and non-contrastive self-supervised approaches. Additionally, we discuss recent theoretical results, augmentation strategies as well as connections to energy-based models. For an exhaustive overview of self-supervised learning, we invite the interested reader to check out two recent surveys (Jing & Tian, 2021; Balestriero et al., 2023).

**Contrastive objectives and connection to mutual information.** Contrastive learning represents an important family of self-supervised learning algorithms, which is rooted in the maximization of mutual information between the data and its latent representation (Linsker, 1988; Becker & Hinton, 1992). Estimating and optimizing mutual information from samples is notoriously difficult (McAllester & Stratos, 2020), especially when dealing with high-dimensional data. Most recent approaches focus on devising variational lower bounds on mutual information (Barber & Agakov, 2004). Indeed, several popular objectives, like the mutual information neural estimation (MINE) (Belghazi et al., 2018), deep InfoMAX (Hjelm et al., 2018), noise

contrastive estimation (InfoNCE) (den Oord et al., 2018; O. Henaff, 2020; Chen et al., 2020; Lee, 2022; Xu et al., 2022) to name a few, all belong to the family of variational lower bounds (Poole et al., 2019). All these estimators have different properties in terms of bias-variance trade-off (Tschannen et al., 2019; Song & Ermon, 2020). Our work model contrastive learning using an equivalent probabilistic graphical model and a corresponding objective function based on the data log-likelihood, thus providing an alternative view to the principle of mutual information maximization. This is similar in spirit to the formulations proposed in the recent works of (Mitrovic et al., 2021; Tomasev et al., 2022; Scherr et al., 2022; Xu et al., 2022). Unlike these works, we are able to extend the log-likelihood perspective to other families of self-supervised approaches and also to highlight and exploit their connections to energy-based models.

**Cluster-based objectives**. There are also recent advances in using clustering techniques in representation learning. For example, DeepCluster (Caron et al., 2018) uses k-means and the produced cluster assignments as pseudo-labels to learn the representation. The work in (Huang et al., 2022) introduces an additional regularizer for deep clustering, invariant to local perturbations applied to the augmented latent representations. The work in (Asano et al., 2020) shows that the pseudo-label assignment can be seen as an instance of the optimal transport problem. SwAV (Caron et al., 2020) proposes to use the Sinkhorn-Knopp algorithm to optimize the optimal transport objective (Cuturi, 2013) and to perform a soft cluster assignment. Finally, contrastive clustering (Li et al., 2021) proposes to minimize the optimal transport objective in a contrastive setting, leveraging both positive and negative samples. Our work provides a simple yet concise formulation of cluster-based self-supervised learning based on the principle of likelihood maximization. Additionally, thanks to the connection with energy-based models, we can perform implicit density estimation and leverage the learnt information to improve the clustering performance.

**Negative-free objectives.** It's important to mention that new self-supervised objectives have recently emerged (Zbontar et al., 2021; Grill et al., 2020) as a way to avoid using negative samples, which are typically required by variational bounds on the mutual information, namely BYOL (Grill et al., 2020), SimSiam (Chen et al., 2020), DINO (Caron et al., 2021), Zero-CL (Zhang et al., 2021), W-MSE (Ermolov et al., 2021), Barlow Twins (Zbontar et al., 2021), VICReg (Bardes et al., 2022a) and its variants (Bardes et al., 2022b), MEC (Liu et al., 2022) and CorInfoMax (Ozsoy et al., 2022). DINO proposes to maximize a cross-entropy objective to match the probabilistic predictions from two augmented versions of the same image. BYOL, SimSiam, W-MSE consider the cosine similarity between the embeddings obtained from the augmented pair of images. Additionally, W-MSE introduces a hard constraint implemented as a differentiable layer to whiten the embeddings. Similarly, Barlow Twins proposes a soft whitening by minimizing the Frobenius norm between the cross-correlation matrix of the embeddings and the identity matrix. Zero-CL pushes forward the idea of whitening the features by also including an instance decorrelation term. VICReg and its variant extend over Barlow Twins by computing the sample covariance matrix, instead of the correlation one (thus avoiding to use batch normalization), by enforcing an identity covariance and by introducing an additional regularizer term to minimize the mean squared error between the embeddings of the two networks and to promote the invariance of the embeddings. Similarly, the work in (Tomasev et al., 2022) uses an invariance loss function in conjunction to the contrastive InfoNCE objective. MEC maximizes the log-determinant of the covariance matrix for the latent representation, thus promoting maximum entropy under the Gaussian assumption. Additionally, CorInfoMax extends MEC by introducing a term that enforces the representation to be invariant under data augmentation. In Section 2.3, we can cast the non-contrastive problem as a minimization of the Kullback Leibler divergence between the latent posterior and a standard normal density prior. In essence, our work allows to compactly represent the family of non-contrastive methods using a likelihood-based objective.

**Additional objectives.** Several works have investigated the relation between different families of self-supervised approaches leading to hybrid objective functions (Kügelgen et al., 2021; Garrido et al., 2023). In contrast, our work attempts to provide a unified view from a probabilistic perspective and to highlight/exploit its connection to energy-based models.

The main idea of generative and discriminative training originally appeared in the context of Bayesian mixture models (Sansone et al., 2016). Specifically, the authors proposed to jointly learn a generative model and cluster data in each class in order to be able to discover subgroups in breast cancer data. Subsequently, the work in (Liu & Abbeel, 2020) pushed the idea forward and apply it to a supervised deep learning

setting. Instead, our work focuses on self-supervised learning and generative models, thus avoiding the need for additional supervision on the categorical variable $y$. Recently, the work in (Li et al., 2022) uses the maximum coding rate criterion to jointly learn an embedding and cluster it. However, the training proceeds in a multistage fashion. In contrast, our work provides a simple formulation enabling to jointly learn and cluster the embeddings in one shot. Additionally, we can leverage the generative perspective to further boost the self-supervised learning performance, as demonstrated by the generation and out-of-distribution detection experiments.

**Theory of self-supervised learning.** Several works have theoretically analysed self-supervised approaches, both for contrastive (Saunshi et al., 2019; Wang & Isola, 2020; Zimmermann et al., 2021; Tosh et al., 2021; HaoChen et al., 2021; Saunshi et al., 2022) and non-contrastive methods (Tian et al., 2021; Kang-Jun et al., 2022; Weng et al., 2022; Wen & Li, 2022), to motivate the reasons for their successes, identify the main underlying principles and subsequently provide more principled/simplified solutions. Regarding the former family of approaches, researchers have (i) identified key properties, such as representation alignment (i.e. feature for positive pairs need to be close to each other) and uniformity (to avoid both representational and dimensional collapse) (Wang & Isola, 2020; Assran, Mahmoud and Caron, Mathilde and Misra, Ishan and Bojanowski, Piotr and Bordes, Florian and Vincent, Pascal and Joulin, Armand and Rabbat, Mike and Ballas, Nicolas, 2022; Assran et al., 2023), (ii) formulated and analyzed the problem using data augmentation graphs (HaoChen et al., 2021) and (iii) examined generalization bounds on the downstream supervised performance (Saunshi et al., 2022; Bao et al., 2022). Regarding the latter family of approaches, the main focus has been devoted to understanding the reasons on why non-contrastive approaches avoid trivial solutions. In this regard, asymmetries, in the form of stop-gradient and diversified predictors, are sufficient to ensure well-behaved training dynamics (Tian et al., 2021; Weng et al., 2022; Wen & Li, 2022). Importantly, the asymmetries are shown to implicitly constrain the optimization during training towards solutions with decorrelated features (Kang-Jun et al., 2022). Recent works have also looked at identifying connections between contrastive and non-contrastive methods (Dubois et al., 2022; Garrido et al., 2023; Balestriero & LeCun, 2022) to unify the two families. The work in (Dubois et al., 2022) proposes a set of desiderata for representation learning, including large dimensional representations, invariance to data augmentations and the use of at least one linear predictor to ensure good performance on linear probe evaluation tasks. The work in (Garrido et al., 2023) analyzes the relations between contrastive and non-contrastive methods, showing their similarities and differences. Both families of approaches learn in a contrastive manner. However, while contrastive solutions learn by contrasting between samples, non-contrastive ones focus on contrasting between the dimensions of the embeddings. The work in (Balestriero & LeCun, 2022) studies the minima in terms of representations for the different loss functions proposed in the two families. The authors are able to show that such minima are equivalent to solutions achieved by spectral methods. This provides additional evidence on the similarities between the approaches and the possibility for their integration.

Our work provides a unifying view of the different classes, allowing to derive several loss functions in a principled manner using variational inference on the data log-likelihood. Additionally, we provide conditions to learn in a principled manner, thus avoiding trivial solutions and the use of asymmetries.

**Generative models and self-supervised learning.** Recently, works have considered synergies between self-supervised and energy-based models (LeCun, 2022) for the purposes of out-of-distribution detection (Hendrycks et al., 2019; Winkens et al., 2020; Mohseni et al., 2020). This is a common characteristics of energy-based models and indeed our work highlight the explicit connection with self-supervised learning. To the best of our knowledge, there is only one recent work exploring the integration between self-supervised learning approaches and energy models (Kim & Ye, 2022). The authors propose to use an energy-based model to learn a joint distribution over the two augmented views for the same data. The resulting objective can be decomposed into a conditional distribution term, leading to a contrastive learning criterion, and a marginal distribution term, leading to an energy-based model criterion. Therefore, the work only considers the integration between contrastive methods and energy-based models. In contrast, our work goes a step forward by showing a general methodology to integrate generative and SSL approaches. Moreover, we provide a new lower bound for the class of cluster-based approaches that guarantees the avoidance of important failure modes.

Table 2: Clustering performance in terms of normalized mutual information (NMI) on test set (moons and circles). Higher values indicate better clustering performance. Mean and standard deviations are computed from 5 different runs.

| Dataset | JEM | GEDI *no prior* | GEDI *no inv* | GEDI *no gen* | GEDI |
|---|---|---|---|---|---|
| Moons | 0.00±0.00 | 0.00±0.00 | 0.11±0.15 | **0.98±0.00** | 0.94±0.07 |
| Circles | 0.00±0.00 | 0.00±0.00 | 0.22±0.13 | 0.83±0.12 | **1.00±0.01** |

There has been also recent interest in combining VAEs with self-supervised learning (Gatopoulos & Tomczak, 2020; Zhue et al., 2020; Wu et al., 2023; Nakamura et al., 2023; Tong et al., 2024). Our formulation is general enough to encompass ELBO-like objectives like the one used in VAEs. Instead of instantiating the generative term by leveraging an energy-based model, one could proceed to lower bound the entropy term following traditional ways to derive an ELBO. We are not interested to pursue this direction, as we aim at devising an objective function which can exploit a simple classifier architecture without the need of additional components, such as a decoder network.

# 6 Experiments

We perform experiments to evaluate the generative/discriminative performance of GEDI and its competitors, namely an energy-based model JEM (Grathwohl et al., 2020), which is trained with persistent contrastive divergence Tieleman (2008) to optimize only the generative term of the GEDI Lower Bound (Eq. (5)), and 2 self-supervised baselines, viz. a negative-free approach based on Barlow Twins (Zbontar et al., 2021) and a cluster-based approach based on SwAV (Caron et al., 2020), which optimize the objectives in Lemma 6 and Lemma 9, respectively. We also compare with a two-stage generative and discriminative solution originally proposed in (Sansone & Manhaeve, 2022), called GEDI *two stage*. The whole analysis is divided into four main experimental settings, the first part provides empirical validation and intuition on the results of Proposition 1 and Theorem 1, the second part based on two synthetic datasets, including moons and circles, compares GEDI against the above-mentioned baselines, the third part extends the comparisons to real-world data, including SVHN, CIFAR-10 and CIFAR-100, and finally the last part showcases the utility of the GEDI framework to mitigate the symbol grounding problem arising in neuro-symbolic learning on MNIST data (Harnad, 1990; Sansone & Manhaeve, 2023b; Marconato et al., 2023). We use existing code both as a basis to build our solution and also to run the experiments for the different baselines. In particular, we use the code from (Duvenaud et al., 2021) for training energy-based models and the repository from (da Costa et al., 2022) for all self-supervised approaches. Our code will be publicly released in its entirety upon acceptance. Implementation details as well as additional experiments are reported in the Supplementary Material.

## 6.1 Part 1: Empirical Validation of the Theory

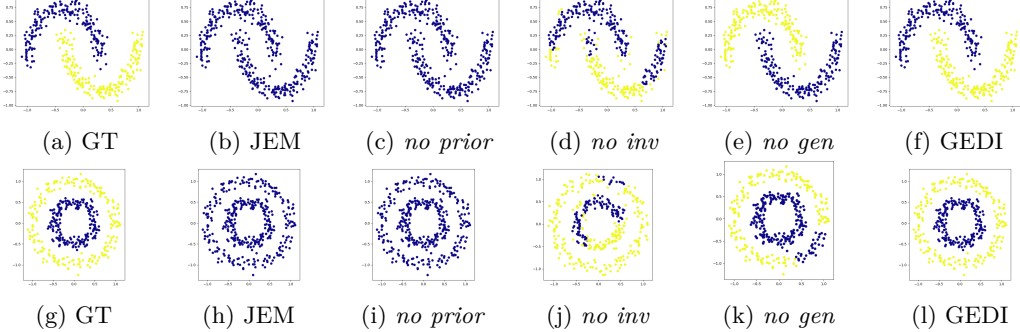

| (a) GT | (b) JEM | (c) *no prior* | (d) *no inv* | (e) *no gen* | (f) GEDI |
|---|---|---|---|---|---|

| (g) GT | (h) JEM | (i) *no prior* | (j) *no inv* | (k) *no gen* | (l) GEDI |
|---|---|---|---|---|---|

Figure 3: Qualitative visualization of the clustering performance for the different strategies on moons (a-f) and on circles (g-l) datasets. Colors identify different cluster predictions. Only GEDI and GEDI *no gen* are able to perform well on both datasets.

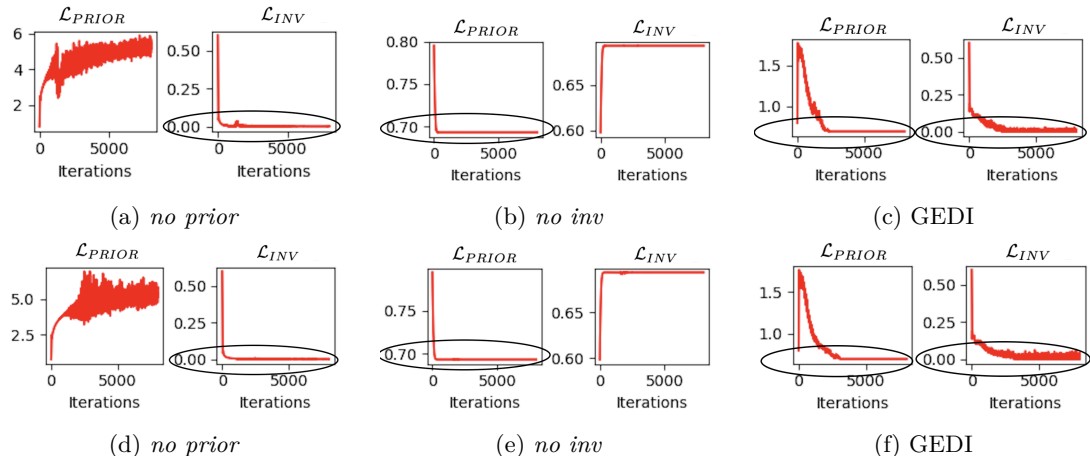

Figure 4: Visualization of the main training losses. As suggested by Eq. 22 in Proposition 1, we can always debug whether the main failure modes in the triad have occurred by checking the minimum value achieved by the losses and compare it with its corresponding global value. Indeed, the global minimum value of each loss is known, that is $\ln(c)$ for $\mathcal{L}_{PRIOR}$ and 0 for $\mathcal{L}_{INV}$. In both moons (a-c) and circles (d-f) the values are $\ln(2) \approx 0.69$ and 0, respectively.

We consider two well-known synthetic datasets, namely moons and circles. We use a multi-layer perceptron (MLP) with two hidden layers (100 neurons each) for *enc* and one with a single hidden layer (4 neurons) for *proj*, we choose $h = 2$ and $\mathcal{T}(x'|x) = N(0, \sigma^2 I)$ with $\sigma = 0.03$ as our data augmentation strategy. We train GEDI for $7k$ iterations using Adam optimizer with learning rate $1e - 3$. Similarly we ablate the contribution of the different loss terms by training the GEDI Lower Bound using only $\mathcal{L}_{GEN}$ (equivalent to JEM), without $\mathcal{L}_{INV}$ (called GEDI *no inv*) without $\mathcal{L}_{PRIOR}$ (called GEDI *no prior*). Further details about the hyperparameters are available in the Supplementary Material (Section G). We evaluate the clustering performance both quantitatively, by using the Normalized Mutual Information (NMI) score and qualitatively, by visualizing the cluster assignments using different colors.

From the results in Table 2 and Figure 3, we can make the following observations: (i) GEDI *no prior* and JEM are subject to cluster collapse on both datasets. This is expected as failure mode 2 is not penalized during training due to the omission of $\mathcal{L}_{PRIOR}$; (ii) GEDI *no inv* is subject to the problem of label inconsistency. Indeed, the obtained cluster labels are not informative and consistent with the underlying manifold structure of the data distribution. Again, this confirms the result of Theorem 1, as failure mode 3 could be avoided by the use of $\mathcal{L}_{INV}$; (iii) GEDI *no gen* achieves competitive performance to GEDI despite the absence of $\mathcal{L}_{GEN}$. While in theory the objective function for this approach admits representation collapse, as predicted by our Theorem, in practice we never observed such issue. It might be the case that the learning dynamics of gradient-based optimisation are enough to avoid the convergence to this trivial solution. Finally (iv) GEDI is guaranteed to avoid the most important failure modes and therefore solve both tasks. We will see later the benefits of including also the generative term in the optimization.

An important consequence of Proposition 1 is that we can use the training losses to debug whether we achieve the global minimum value and therefore avoid the triad of failure modes. Figure 4 showcases this property.

## 6.2 Part 2: Generative/Discriminative Comparisons on Synthetic Data

We consider the same experimental setting used in part 1 and train JEM, SwAV and GEDI for $7k$ iterations using Adam optimizer with learning rate $1e - 3$. Further details about the hyperparameters are available in the Supplementary Material (Section G). We evaluate the clustering performance both qualitatively, by visualizing the cluster assignments using different colors, and quantitatively, by using the Normalized

Table 3: Clustering performance in terms of normalized mutual information (NMI) on test set (moons and circles). Higher values indicate better clustering performance. Mean and standard deviations are computed from 5 different runs.

| Dataset | JEM | Barlow | SwAV | GEDI *no gen* | GEDI | Gain |
|---|---|---|---|---|---|---|
| Moons | 0.00±0.00 | 0.22±0.10 | 0.76±0.36 | **0.98±0.00** | 0.94±0.07 | **+0.22** |
| Circles | 0.00±0.00 | 0.13±0.10 | 0.00±0.00 | 0.83±0.12 | **1.00±0.01** | **+0.87** |

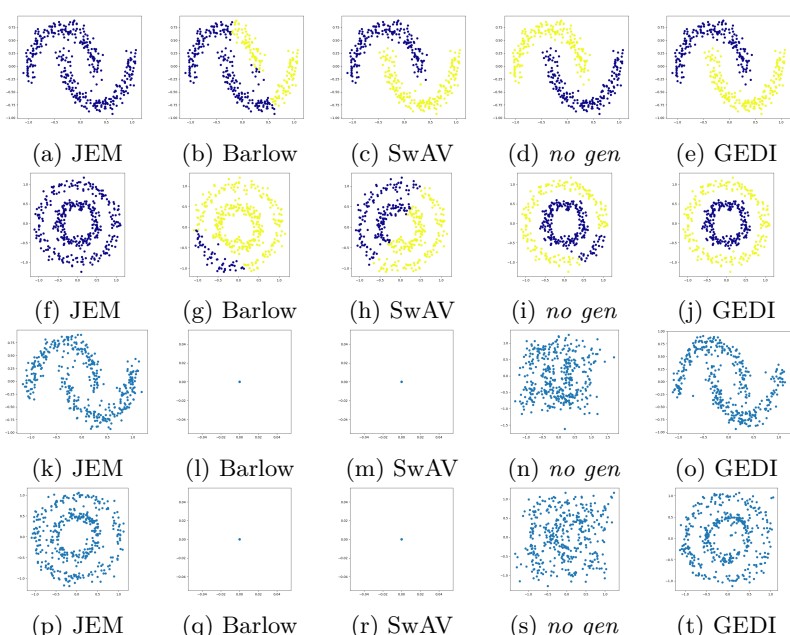

|  |  |  |  |  |
|---|---|---|---|---|
| (a) JEM | (b) Barlow | (c) SwAV | (d) *no gen* | (e) GEDI |
| (f) JEM | (g) Barlow | (h) SwAV | (i) *no gen* | (j) GEDI |
| (k) JEM | (l) Barlow | (m) SwAV | (n) *no gen* | (o) GEDI |
| (p) JEM | (q) Barlow | (r) SwAV | (s) *no gen* | (t) GEDI |

Figure 5: Qualitative visualization of the clustering performance for the different strategies on moons (a-e) and on circles (f-j) datasets. Colors identify different cluster predictions. Only GEDI and GEDI *no gen* are able to perform well on both datasets. Qualitative visualization of the generative performance for the different strategies on moons (k-o) and on circles (p-t) datasets. Colors identify different cluster predictions. All GEDI approaches (except for no gen perform comparably well to the generative solution JEM).

Mutual Information (NMI) score. Additionally, we qualitatively demonstrate the generative performance of all approaches.

We report all quantitative performance in Table 3. As expected, Barlow fails to solve both tasks due to the different nature to cluster-based approaches. SwAV correctly solves the task for the moons dataset, while it faces difficulty to solve the one based on circles. This is due to the fact that the optimized bound in Eq. (17) includes several trivial solutions. The introduced asymmetries (such as stop gradient and the clustering layers) are not enough to rule them out. This is confirmed by visually inspecting the latent representation and observing that the encoder collapses to the identity function. Therefore, the projector head can solve the task by simply learning a linear separator. As a consequence, we observe that half of the labels are permuted across the two manifolds (refer to Figure 5). In contrast, GEDI can recover the true clusters in both datasets without any additional asymmetry and have a guarantee to avoid the triad of failure modes. Moreover, GEDI is able to learn a correct density estimator compared to the SSL baselines (cf. Figure 5). This is an important property that can come in handy for out-of-distribution detection, as we will see later in the real-world experiments.

Table 4: Generative and discriminative performance on test set (SVHN, CIFAR-10, CIFAR-100). Normalized mutual information (NMI) and Frechet Inception Distance (FID) are used as evaluation metrics for the discriminative and generative tasks, respectively. Higher values of NMI and lower values of FID indicate better performance. Mean and standard deviations are reported for 5 different initialization seeds.

| Task | Method | SVHN | CIFAR-10 | CIFAR-100 |
|---|---|---|---|---|
| Discriminative (NMI) | JEM | 0.00±0.00 | 0.00±0.00 | 0.00±0.00 |
| | Barlow | 0.20±0.02 | 0.17±0.04 | 0.61±0.05 |
| | SwAV | 0.21±0.01 | 0.44±0.01 | 0.51±0.21 |
| | GEDI *no gen* | **0.27±0.04** | **0.45±0.00** | **0.87±0.00** |
| | GEDI | **0.25±0.04** | 0.44±0.01 | **0.87±0.00** |
| Generative (FID) | JEM | 201±36 | 223±15 | 271±85 |
| | Barlow | 334±28 | 382±22 | 403±28 |
| | SwAV | 480±67 | 410±31 | 420±26 |
| | GEDI *no gen* | 488±43 | 403±9 | 435±11 |
| | GEDI | 193±10 | **214±13** | **226±9** |

Table 5: OOD detection in terms of AUROC on test set (CIFAR-10, CIFAR-100). Training is performed on SVHN.

| Dataset | JEM | Barlow | SwAV | GEDI *no gen* | GEDI |
|---|---|---|---|---|---|
| CIFAR-10 | 0.73 | 0.17 | 0.26 | 0.1 | **0.80** |
| CIFAR-100 | 0.72 | 0.24 | 0.32 | 0.15 | **0.80** |

### 6.3 Part 3: Real-world Experiments on SVHN, CIFAR-10, CIFAR-100

We consider three well-known computer vision benchmarks, namely SVHN, CIFAR-10 and CIFAR-100. We use a simple 8-layer Resnet network for the backbone encoder for both SVHN and CIFAR-10 (around 1M parameters) and increase the hidden layer size for CIFAR-100 (around 4.1M parameters) as from (Duvenaud et al., 2021). We use a MLP with a single hidden layer for *proj* (the number of hidden neurons is double the number of inputs), we choose $h = 256$ for CIFAR-100 and $h = 128$ for all other cases. Additionally, we use data augmentation strategies commonly used in the SSL literature, including color jitter, and gray scale conversion to name a few. We train JEM, Barlow, SwAV, GEDI *no gen* and GEDI using Adam optimizer with learning rate $1e-4$ and batch size 64 for 20, 200 and 200 epochs for each respective dataset (SVHN, CIFAR-10 AND CIFAR-100). Further details about the hyperparameters are available in the Supplementary Material (Section I). Similarly to the toy experiments, we evaluate the clustering performance by using the Normalized Mutual Information (NMI) score. Additionally, we evaluate the generative performance qualitatively using

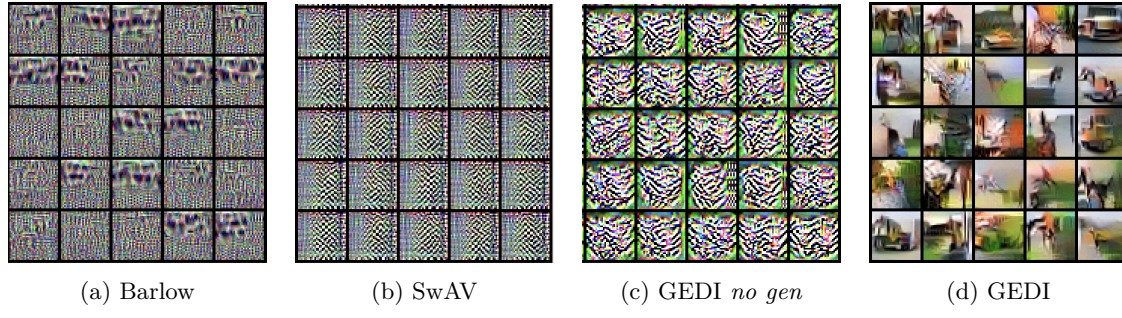

(a) Barlow      (b) SwAV      (c) GEDI *no gen*      (d) GEDI

Figure 6: Samples generated by the different discriminative strategies on CIFAR-10 running Stochastic Langevin Dynamics for 500 iterations.

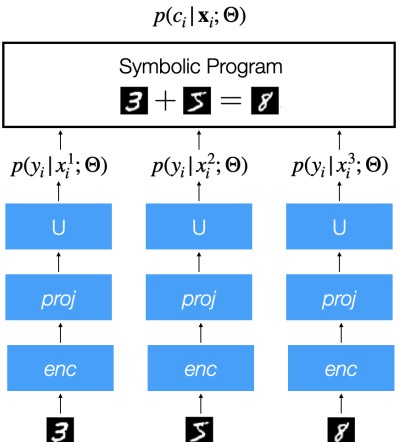

Figure 7: The full model for the neurosymbolic experiment using GEDI.

the Frechet Inception Distance (Heusel et al., 2017) as well as the OOD detection capabilities following the methodology in (Grathwohl et al., 2020).

From Table 4, we observe that GEDI is able to outperform all other competitors by a large margin, thanks to the properties of both generative and self-supervised models. We observe that the difference gap in clustering performance increases with CIFAR100. This is due to a larger size of the backbone used in the CIFAR100 experiments (cf. the size of the latent representation). In terms of generation performance, GEDI is the only approach that compares favorably with JEM. We provide a qualitative set of samples generated by the different discriminative models in Figure 6. Last but not least, we investigate the OOD detection capabilities of the different methods. Table 11 provides a quantitative summary of the performance for a subset of experiments (the complete set is available in Section J). We observe that GEDI is more robust compared to other discriminative baselines, thanks to its generative nature. Overall, these experiments provide real-world evidence on the benefits of the proposed unification and theoretical results.

### 6.4 Part 4: Tackling A Non-Trivial Instantiation of the Symbol Grounding Problem

For the final task, we consider applying the self-supervised learning approach to a neuro-symbolic (NeSy) setting. For this, we borrow an experiment from DeepProbLog (Manhaeve et al., 2018; 2021). In this task, each example consists of a three MNIST images such that the value of the last one is the sum of the first two, e.g. $\boxed{3} + \boxed{5} = \boxed{8}$. This can thus be considered a minimal neuro-symbolic tasks, as it requires a minimal reasoning task (a single addition) on top of the image classification task. This task only contains positive examples. We use the neural network from Manhaeve et al. (2018) as the neural backbone. We optimize $\mathcal{L}_{NESY}$ instead of $\mathcal{L}_{INV}$, as the NeSy loss also forces a clustering of the digits, but is more informed. To calculate $\mathcal{L}_{NESY}$, we use GEDI to classify the images to produce a probability distribution over the classes 0 to 9, and we use the inference mechanism from DeepProbLog to calculate the probability that the sum holds as shown in Figure 7. For this setting, this coincides with the Semantic Loss function (Xu et al., 2018). The NeSy constraint loss is defined as:

$$\mathcal{L}_{NESY}(\Theta) = \sum_{i=1}^{n} CE\left(1, p(c|\mathbf{x}_i; \Theta)\right)$$

where $\mathbf{x}_i$ is now an arbitrary number of images in each data point. For this experiment, these are the three MNIST images $\mathbf{x}_i = (x_i^1, x_i^2, x_i^3)$. The probability of the constraint $p(c|\mathbf{x}_i; \Theta)$ is defined in terms of a set of rules $\mathcal{R}$ and a set of probabilistic facts $\mathcal{F} = \{f_1 \ldots f_n\}$ with probabilities $p(f_j|\mathbf{x}_i; \Theta)$. For this setting, there are 30 probabilistic facts, one for each possible classifications of each digit, with their probabilities is thus parameterized by the neural network. The probability of the facts is thus

$$p(y_i = 0|x_i^1; \Theta), \ldots, p(y_i = 9|x_i^1; \Theta), p(y_i = 0|x_i^2; \Theta), \ldots, p(y_i = 9|x_i^3; \Theta)$$

.

Each subset $F \subset \mathcal{F}$ defines a possible world $w_F = \{a \mid \mathcal{R} \cup F \models a\}$, i.e. the set of all atoms $a$ entailed by the rules and the facts in $F$. In this setting, this is a total assignment of classes to digits, and the truth value of the constraint. The probability of the constraint is the sum of the probability of each possible world (i.e. classifications of the digits) in which the constraint holds.

$$P(c_i|\mathbf{x}_i;\Theta) = \sum_{F \mid c \in w_F} P(w_F|\mathbf{x}_i;\Theta) \tag{8}$$

Where the probability of the possible world is defined as the product of the probability of each fact that is true in this possible word, and one minus the probability of each fact that does not hold in the possible world.

$$P(w_F|\mathbf{x}_i;\Theta) = \prod_{f_j \in F} p(f_j|\mathbf{x}_i;\Theta) \prod_{f_j \in \mathcal{F} \setminus F} (1 - p(f_j|\mathbf{x}_i;\Theta)) \tag{9}$$

It is a specifically interesting use case for representation learning, since when only the constraint probability is optimized, the neural network tends to collapse onto the trivial solution of classifying each digit as a 0, as shown in (Manhaeve et al., 2018; 2021; Sansone & Manhaeve, 2023a;b). This is a logically correct solution, but an undesirable solution. Optimizing the uniformity objective should prevent this collapse. A neural network should be able to correctly classify MNIST digits by using the SSL training objective and the logical constraint. Here, the focus is on the small data regime, and see whether the logical constraint is able to provide additional information. We also look at the performance of SwAV when combined with the logical constraint. Further details about the hyperparameters and experimental setup are available in Appendix M. We evaluate the model by measuring the accuracy and NMI of the model on the MNIST test dataset for different numbers of training examples. The results are shown in Table 6. Here, # Ex. examples indicates the number of addition examples, which each have 3 MNIST digits. The results show that, without the NeSy constraint, the mean accuracy is low for all settings. The NMI is higher, however, and increases as there is more data available. This is expected, since the model is able to learn how to cluster from the data, but unable to learn how to classify. By including the constraint loss, the accuracy improves, as the model now has information on which cluster belongs to which class. Furthermore, it also has a positive effect on the NMI, as we have additional information on the clustering which is used by the model. Combining SwAV with the logical constraint severely underperforms. It does not sufficiently enforce the uniformity of the predicted labels, which leads to the a local optimum where the predictions are distributed over 2-3 classes. These results show us that the proposed method is beneficial to learn to correctly recognize MNIST images using only a weakly-supervised constraint. Furthermore, we show that the proposed method can leverage the constraint to further improve NMI and classification accuracy.

Table 6: The median and standard deviation of the accuracy and NMI of GEDI and SwAV on the MNIST test set after training on the addition dataset. We also include results for learning without GEDI (only optimizing the constraint), and training GEDI without the NeSy constraint.

| | No GEDI | | Only GEDI | | SwAV & NeSy | | GEDI & NeSy | |
|---|---|---|---|---|---|---|---|---|
| # Ex. | Acc. | NMI | Acc. | NMI | Acc. | NMI | Acc. | NMI |
| 100 | $.10 \pm .00$ | $.00 \pm .00$ | $.08 \pm .01$ | $.22 \pm .03$ | $.14 \pm .00$ | $.19 \pm .00$ | $\mathbf{.42 \pm .02}$ | $\mathbf{.35 \pm .01}$ |
| 1000 | $.10 \pm .00$ | $.00 \pm .00$ | $.09 \pm .03$ | $.29 \pm .06$ | $.10 \pm .00$ | $.36 \pm .00$ | $\mathbf{.96 \pm .01}$ | $\mathbf{.91 \pm .01}$ |
| 10000 | $.10 \pm .00$ | $.00 \pm .00$ | $.11 \pm .05$ | $.44 \pm .10$ | $.10 \pm .00$ | $.40 \pm .00$ | $\mathbf{.97 \pm .00}$ | $\mathbf{.93 \pm .00}$ |

# 7 Conclusions and Future Research

We have presented a unified perspective on self-supervised clustering and energy-based models. The corresponding GEDI lower bound is guaranteed to prevent three main failure modes: representation collapse, cluster collapse, and label inconsistency of cluster assignments. By shedding new light on the synergies

between self-supervised learning and likelihood-based generative models, we aim to inspire subsequent studies proposing new implementations of our general methodology, such as novel approaches to integrating self-supervised learning and latent variable models. Additionally, we demonstrate that GEDI can better capture the underlying data manifolds and provide more precise predictions compared to existing strategies. We believe that further progress can be achieved by combining self-supervised learning with other areas of mathematics, such as topology and differential geometry, for instance to guarantee a notion of connectedness when dealing with the manifold assumption. Moreover, we show that GEDI can be easily integrated into existing statistical relational reasoning frameworks, opening the door to new neuro-symbolic integrations and enabling the handling of low data regimes, which are currently beyond the reach of existing self-supervised learning solutions. There are several area of improvements and extension that we will target. First and foremost, we plan in the near future to apply the existing solution to larger scale datasets, such as Imagenet. Secondly, the current solution assumes that information about the number of classes and the class prior is known in advance. This might limit the applicability of the proposed solution to real-world cases, such as open-world settings or scenarios with long tailed distributions. It will be certainly interesting to generalize the framework to deal with such new settings. Finally, it will be possible to extend the proposed framework towards more object-centric representation learning, thus going beyond the traditional object classification problem.

## 8 Author Contributions and Acknowledgements

The authors would like to thank Michael Puthawala for initial discussion on GEDI. ES had the idea, developed the theory around the three key challenges, implemented the solution for the toy and real cases and wrote the paper. RM implemented and wrote the experiments for the neuro-symbolic setting and also helped by proof-reading the article. This work received funding from the Horizon Europe research and innovation programme (MSCA-GF grant agreement n° 101149800) and also from the Flemish Government under the "Onderzoeksprogramma Artificiele Intelligentie (AI) Vlaanderen" programme. The computational resources and services used in this work were provided by the computing infrastructure in the Electrical Engineering Department (PSI group) and the Department of Computer Science (DTAI group) at KU Leuven.

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

# A    Appendix

# B    Proof of Lemma 3

**Definition 11** (Restated). *Define the ground truth joint distribution for the graphical model in Fig. 2(a) as $p(x_{1:n}, y_{1:n}) \equiv \prod_{j=1}^{n} p(x_j)\delta(y_j - j)$ with $y_j \in \{1, \ldots, n\}$ and $\delta$ a delta function. Moreover, define the model distribution as $p(x_{1:n}, y_{1:n}; \Theta) \equiv \prod_{i=1}^{n} p(x_j)p(y_i|x_i; \Theta)$ with $p(y_i|x_i; \Theta) \equiv \frac{e^{sim(g(x_i), g(x_{y_i}))/\tau}}{\sum_{k=1}^{n} e^{sim(g(x_i), g(x_k))/\tau}}$, $sim : \mathbb{R}^l \times \mathbb{R}^l \to \mathbb{R}$ being a similarity function, $\tau > 0$ a temperature parameter and $\Theta = \{\theta, \{x_i\}_i^n\}$ a set of parameters.*

**Lemma 4** (Restated). *Given Definition 11, maximizing the InfoNCE objective (den Oord et al., 2018) is equivalent to maximize the following log-likelihood objective:*

$$E_{p(x_{1:n}, y_{1:n})}\{\log p(x_{1:n}, y_{1:n})\} \geq \underbrace{-H_p(x_{1:n})}_{Neg.\ entropy\ term} + \underbrace{\mathbb{E}_{\prod_{j=1}^{n} p(x_j)}\left\{\sum_{i=1}^{n} \log \frac{e^{sim(g(x_i), g(x_i))/\tau}}{\sum_{k=1}^{n} e^{sim(g(x_i), g(x_k))/\tau}}\right\}}_{Predictive\ term\ \mathcal{L}_{CT}(\Theta)} \quad (10)$$

*Moreover, the maximization of this lower bound is equivalent to solve an instance classification problem.*

*Proof.* We first recall the InfoNCE objective (cf. Eq. 10 in (Poole et al., 2019))

$$\mathcal{I}_{NCE} \equiv E_{\prod_{j=1}^n p(x_j)\delta(z_i - g(x_j))} \left\{ \frac{1}{n} \sum_{i=1}^n \log \frac{e^{f(x_i,z_i)}}{\frac{1}{n}\sum_{k=1}^n e^{f(x_i,z_k)}} \right\} \tag{11}$$

This objective can be rewritten in the following way:

$$\mathcal{I}_{NCE} = \frac{1}{n} E_{\prod_{j=1}^n p(x_j)\delta(z_i - g(x_j))} \left\{ \sum_{i=1}^n \log \frac{e^{f(x_i,z_i)}}{\sum_{k=1}^n e^{f(x_i,z_k)}} \right\} + log(n)$$

$$\propto E_{\prod_{j=1}^n p(x_j)\delta(z_i - g(x_j))} \left\{ \sum_{i=1}^n \log \frac{e^{f(x_i,z_i)}}{\sum_{k=1}^n e^{f(x_i,z_k)}} \right\}$$

$$= E_{\prod_{j=1}^n p(x_j)} \left\{ \sum_{i=1}^n \log \frac{e^{f(x_i,g(x_i))}}{\sum_{k=1}^n e^{f(x_i,g(x_k))}} \right\} \tag{12}$$

Next, we highlight an important relation between the ground truth and model log-likelihoods:

$$E_{p(x_{1:n},y_{1:n})}\{\log p(x_{1:n}, y_{1:n})\} = E_{p(x_{1:n},y_{1:n})}\{\log p(x_{1:n}, y_{1:n}; \Theta)\} + KL\{p(x_{1:n}, y_{1:n})\|p(x_{1:n}, y_{1:n}; \Theta)\}$$
$$\geq E_{p(x_{1:n},y_{1:n})}\{\log p(x_{1:n}, y_{1:n}; \Theta)\} \tag{13}$$

In other words, maximizing the term $E_{p(x_{1:n},y_{1:n})}\{p(x_{1:n}, y_{1:n}; \Theta)\}$ is equivalent to minimize the $KL$ term, thus solving an instance discrimination problem, where the model matches the instance classifier. This proves the last part of the lemma.

Now, we are left to show the first part. We can elaborate the above lower bound even further by using the definitions at the basis of the Lemma. Indeed,

$$E_{p(x_{1:n},y_{1:n})}\{\log p(x_{1:n}, y_{1:n}; \Theta)\} = E_{\prod_{j=1}^n p(x_j)\delta(y_j - j)} \left\{ \log \prod_{i=1}^n p(x_i)p(y_i|x_i; \Theta) \right\}$$

$$= \sum_{i=1}^n E_{\prod_{j=1}^n p(x_j)\delta(y_j - j)}\{\log p(x_i)p(y_i|x_i; \Theta)\}$$

$$= \sum_{i=1}^n E_{p(x_i)\delta(y_i - i)}\{\log p(x_i)p(y_i|x_i; \Theta)\}$$

$$= \sum_{i=1}^n E_{p(x_i)}\{\log p(x_i)p(y_i = i|x_i; \Theta)\}$$

$$= \sum_{i=1}^n E_{p(x_i)}\{\log p(x_i)\} + \sum_{i=1}^n E_{p(x_i)}\{\log p(y_i = i|x_i; \Theta)\}$$

$$= -H_p(x_{1:n}) + \sum_{i=1}^n E_{p(x_i)}\{\log p(y_i = i|x_i; \Theta)\}$$

$$= -H_p(x_{1:n}) + E_{\prod_{j=1}^n p(x_j)} \left\{ \sum_{i=1}^n \log p(y_i = i|x_i; \Theta) \right\}$$

$$= -H_p(x_{1:n}) + \mathbb{E}_{\prod_{j=1}^n p(x_j)} \left\{ \sum_{i=1}^n \log \frac{e^{sim(g(x_i),g(x_i))/\tau}}{\sum_{k=1}^n e^{sim(g(x_i),g(x_k))/\tau}} \right\} \tag{14}$$

Now, by defining $f(x, z) \equiv sim(x, z)/\tau$ and substituting it into Eq. 14, we can match with Eq. 12, thus concluding the proof. $\square$

## C  Proof of Lemma 6

**Definition 12** (Restated). *Define the ground truth joint distribution for the graphical model in Fig. 2(b) as $p(z, \xi_{1:n}, x_{1:n}, x'_{1:n}) \equiv p(z) \prod_{i=1}^n p(x_i|z)p(\xi_i)p(x'_i|x_i, \xi_i)$ with Gaussian priors $p(z) = \mathcal{N}(z|0, I)$, $p(\xi_i) = \mathcal{N}(\xi_i|0, \gamma^{-1}I)$ and assume the following conditional independencies $p(x'_i|x_i, \xi_i) = \mathcal{T}(x'_i|x_i)$ and $p(x_i|z) = p(x_i)$. Moreover, introduce the following auxiliary model distributions, that is $q(\xi_i|x_i, x'_i) \equiv \mathcal{N}(\xi_i|enc(x_i) - enc(x'_i), I)$ and $q(z|x_{1:n}) \equiv \mathcal{N}(z|0, \Sigma)$, with $\Sigma \equiv \sum_{i=1}^n (g(x_i) - \bar{g})(g(x_i) - \bar{g})^T + \beta I$, $\beta > 0$ chosen to ensure the positive definiteness of $\Sigma$ and $\bar{g} = 1/n \sum_{i=1}^n g(x)_i$. Finally, define $\Theta = \{\theta\}$ as the set of parameters.*

**Lemma 5** (Restated). *Given Definition 12, maximizing the CorInfoMax objective (Ozsoy et al., 2022) is equivalent to maximize the following log-likelihood lower bound:*

$$E_{p(x_{1:n}, x'_{1:n})}\{\log p(x_{1:n}, x'_{1:n})\} \geq \underbrace{-H_p(x_{1:n})}_{Neg.\ entropy\ term} \underbrace{-E_{p(x_{1:n})}\{KL(q(z|x_{1:n})\|p(z))\}}_{discriminative\ SSL\ term\ \mathcal{L}_{NF}(\Theta)}$$

$$\underbrace{-\sum_{i=1}^n E_{p(x_i)\mathcal{T}(x'_i|x_i)}\{KL(q(\xi_i|x_i, x'_i)\|p(\xi_i))\}}_{Continuation\ of\ \mathcal{L}_{NF}(\Theta)} + const$$

*where the first Kullback-Leibler term $KL(q(z|x_{1:n})\|p(z)) \equiv KL(\mathcal{N}(z|0, \Sigma)\|\mathcal{N}(z|0, I))$, the second term $KL(q(\xi_i|x_i, x'_i)\|p(\xi_i)) \equiv KL(\mathcal{N}(\xi_i|enc(x_i) - enc(x'_i), I)\|\mathcal{N}(\xi_i|0, \gamma^{-1}I))$ and const being a constant for the optimization over $\theta$.*

*Proof.* We first recall the CorInfoMax objective (cf Eq. 6 in (Ozsoy et al., 2022))

$$\mathcal{L}_{CorInfoMax} \equiv E_{p(x_{1:n})}\{\log|\Sigma|\} + \gamma \sum_{i=1}^n E_{p(x_i)\mathcal{T}(x'_i|x_i)}\{\|g(x_i) - g(x'_i)\|^2\} \tag{15}$$

with $\gamma$ a positive scalar to weight the two objective terms.

Next, we derive the log-likelihood lower bound in the Lemma.

$$E_{p(x_{1:n}, x'_{1:n})}\{\log p(x_{1:n}, x'_{1:n})\} = E_{p(x_{1:n}, x'_{1:n})}\left\{\log \int_z \int_{\xi_{1:n}} p(z) \prod_{i=1}^n p(x_i|z)p(\xi_i)p(x'_i|x_i, \xi_i)\right\}$$

$$= E_{p(x_{1:n}, x'_{1:n})}\left\{\log \int_z \int_{\xi_{1:n}} p(z) \prod_{i=1}^n p(x_i)p(\xi_i)\mathcal{T}(x'_i|x_i)\right\}$$

$$= E_{p(x_{1:n}, x'_{1:n})}\left\{\log \prod_{i=1}^n p(x_i)\mathcal{T}(x'_i|x_i)\right\} + E_{p(x_{1:n})}\left\{\log \int_z p(z)\right\}$$

$$+ E_{p(x_{1:n}, x'_{1:n})}\left\{\log \int_{\xi_{1:n}} \prod_{i=1}^n p(\xi_i)\right\}$$

$$= -H_p(x_{1:n}) + const + E_{p(x_{1:n})}\left\{\log \int_z p(z)\right\}$$

$$+ E_{p(x_{1:n}, x'_{1:n})}\left\{\log \int_{\xi_{1:n}} \prod_{i=1}^n p(\xi_i)\right\}$$

$$= -H_p(x_{1:n}) + const + E_{p(x_{1:n})}\left\{\log \int_z q(z|x_{1:n})\frac{p(z)}{q(z|x_{1:n})}\right\}$$

$$+ E_{p(x_{1:n}, x'_{1:n})}\left\{\log \prod_{i=1}^n \int_{\xi_i} q(\xi_i|x_i, x'_i)\frac{p(\xi_i)}{q(\xi_i|x_i, x'_i)}\right\}$$

$$\geq -H_p(x_{1:n}) + const + E_{p(x_{1:n})}\left\{\int_z q(z|x_{1:n})\log \frac{p(z)}{q(z|x_{1:n})}\right\}$$

$$+ \prod_{i=1}^{n} E_{p(x_i)\mathcal{T}(x_i'|x_i)} \left\{ \int_{\xi_i} q(\xi_i|x_i, x_i') \log \frac{p(\xi_i)}{q(\xi_i|x_i, x_i')} \right\}$$

$$\geq -H_p(x_{1:n}) + \text{const} + E_{p(x_{1:n})} \left\{ \int_z q(z|x_{1:n}) \log \frac{p(z)}{q(z|x_{1:n})} \right\}$$

$$+ \prod_{i=1}^{n} E_{p(x_i)\mathcal{T}(x_i'|x_i)} \left\{ \int_{\xi_i} q(\xi_i|x_i, x_i') \log \frac{p(\xi_i)}{q(\xi_i|x_i, x_i')} \right\}$$

$$= -H_p(x_{1:n}) - E_{p(x_{1:n})}\{KL(q(z|x_{1:n})\|p(z))\}$$

$$- \sum_{i=1}^{n} E_{p(x_i)\mathcal{T}(x_i'|x_i)}\{KL(q(\xi_i|x_i, x_i')\|p(\xi_i))\} + \text{const} \tag{16}$$

Now, observe that the first KL term can be rewritten as follows:

$$E_{p(x_{1:n})}\{KL(q(z|x_{1:n})\|p(z))\} = KL(\mathcal{N}(z|0, \Sigma)\|\mathcal{N}(z|0, I))$$

$$= E_{p(x_{1:n})} \left\{ \frac{Tr(\Sigma)}{2} - \frac{\log |\Sigma|}{2} \right\} + \text{const}'$$

$$= \frac{1}{2} E_{p(x_{1:n})}\{\log |\Sigma|\} + \text{const}''$$

where the last quality holds whenever $g$ has a batch normalization layer in its output, thus making $Tr(\Sigma)$ a constant for the optimization.

The second KL term can be rewritten as follows:

$$\sum_{i=1}^{n} E_{p(x_i)\mathcal{T}(x_i'|x_i)}\{KL(q(\xi_i|x_i, x_i')\|p(\xi_i))\} =$$

$$= \sum_{i=1}^{n} E_{p(x_i)\mathcal{T}(x_i'|x_i)}\{KL(\mathcal{N}(\xi_i|enc(x_i) - enc(x_i'), I)\|\mathcal{N}(\xi_i|0, \gamma^{-1}I))\}$$

$$= \frac{\gamma}{2} \sum_{i=1}^{n} E_{p(x_i)\mathcal{T}(x_i'|x_i)}\{\|enc(x_i) - enc(x_i')\|^2\} + \text{const}'''$$

Substituting both expresssions into Eq. 16 and choosing $g \equiv enc$, we recover the CorInfoMax objective up to a factor $1/2$ and an additive constant. $\qquad\square$

## D  Proof of Lemma 9

**Definition 13** (Restated). *Define the ground truth joint distribution for the graphical model in Fig. 2(c) as $p(x_{1:n}, x_{1:n}', y_{1:n}) \equiv \prod_{i=1}^{n} p(x_i)\mathcal{T}(x_i'|x_i)p(y_i|x_i; \Theta)$ with $y_i \in \{1, \ldots, c\}$ being a categorical variable to identify one of $c$ clusters, $p(y_i|x_i; \Theta) = \frac{e^{U_{:y_i}^T G_{:i}/\tau}}{\sum_y e^{U_{:y}^T G_{:i}/\tau}}$, where $U \in \mathbb{R}^{h \times c}$ is the matrix[6] of cluster centers, $G = [g(x_1), \ldots, g(x_n)] \in \mathbb{R}^{h \times n}$ is a representation matrix and $\Theta = \{\theta, U\}$ is the set of parameters. Moreover, introduce the auxiliary clustering distribution $q(y_{1:n}|x_{1:n}') \equiv \prod_{i=1}^{n} q(y_i|x_i')$.*

We can state the following Lemma (proof is provided in Appendix D)

**Lemma 6** (Restated). *Given Definition 13, maximizing the SwAV objective (Caron et al., 2020) is equivalent to maximize the following log-likelihood lower bound:*

$$E_{p(x_{1:n}, x_{1:n}')}\{\log p(x_{1:n}, x_{1:n}')\} \geq \underbrace{-H_p(x_{1:n})}_{\text{Neg. entropy term}} + const$$

---

[6]We use subscripts to select rows and columns. For instance, $U_{:y}$ identify $y-$th column of matrix $U$.

$$+ \underbrace{\sum_{i=1}^{n} \mathbb{E}_{p(x_i)\mathcal{T}(x_i'|x_i)} \left\{ \mathbb{E}_{q(y_i|x_i')} \log p(y_i|x_i; \Theta) + H_q(y_i|x_i') \right\}}_{\text{discriminative SSL term } \mathcal{L}_{CB}(\Theta)}$$

*Proof.* We first recall the SwAV objective (cf. Eq. 2 and 3 in (Caron et al., 2020))

$$\mathcal{L}_{\text{SwAV}}(\Theta) = \sum_{i=1}^{n} \mathbb{E}_{p(x_i)\mathcal{T}(x_i'|x_i)} \{ \mathbb{E}_{q(y_i|x_i')} \log p(y_i|x_i; \Theta) \}$$

$$\mathcal{L}_{\text{SwAV}}(Q) = \mathbb{E}_{p(x_{1:n}, x_{1:n}')} \{ Tr(QU^T G) \} + \tau \mathbb{E}_{p(x_{1:n}, x_{1:n}')} \{ H_Q(y_{1:n}|x_{1:n}') \}$$

where where $Q \equiv [q(y_1|x_1'), \ldots, q(y_n|x_n')]^T$ is a prediction matrix of size $n \times c$ and $Tr(\cdot)$ is the trace operator for any input matrix.

Next, we derive the log-likelihood lower bound:

$$E_{p(x_{1:n}, x_{1:n}')} \{ \log p(x_{1:n}, x_{1:n}') \} = E_{p(x_{1:n}, x_{1:n}')} \left\{ \log \sum_{y_{1:n}} \prod_{i=1}^{n} p(x_i)\mathcal{T}(x_i'|x_i)p(y_i|x_i; \Theta) \right\}$$

$$= E_{p(x_{1:n}, x_{1:n}')} \left\{ \log \prod_{i=1}^{n} \sum_{y_i} p(x_i)\mathcal{T}(x_i'|x_i)p(y_i|x_i; \Theta) \right\}$$

$$= -H_p(x_{1:n}) + \text{const}$$

$$+ \sum_{i=1}^{n} E_{p(x_i)\mathcal{T}(x_i'|x_i)} \left\{ \log \sum_{y_i} p(y_i|x_i; \Theta) \right\}$$

$$= -H_p(x_{1:n}) + \text{const}$$

$$+ \sum_{i=1}^{n} E_{p(x_i)\mathcal{T}(x_i'|x_i)} \left\{ \log \sum_{y_i} q(y_i|x_i') \frac{p(y_i|x_i; \Theta)}{q(y_i|x_i')} \right\}$$

$$\geq -H_p(x_{1:n}) + \text{const}$$

$$+ \sum_{i=1}^{n} E_{p(x_i)\mathcal{T}(x_i'|x_i)} \left\{ \sum_{y_i} q(y_i|x_i') \log \frac{p(y_i|x_i; \Theta)}{q(y_i|x_i')} \right\}$$

$$= -H_p(x_{1:n}) + \text{const}$$

$$+ \sum_{i=1}^{n} \mathbb{E}_{p(x_i)\mathcal{T}(x_i'|x_i)} \left\{ \mathbb{E}_{q(y_i|x_i')} \log p(y_i|x_i; \Theta) + H_q(y_i|x_i') \right\}$$

We observe that the maximization of this lower bound with respect to $\Theta$ is equivalent to maximize $\mathcal{L}_{SwAV}(\Theta)$. We can also show the equivalence to $\mathcal{L}_{SwAV}(Q)$ by expressing the lower bound in vector format, namely:

$$\sum_{i=1}^{n} \mathbb{E}_{p(x_i)\mathcal{T}(x_i'|x_i)} \left\{ \mathbb{E}_{q(y_i|x_i')} \log p(y_i|x_i; \Theta) + H_q(y_i|x_i') \right\} =$$

$$= \sum_{i=1}^{n} \mathbb{E}_{p(x_i)\mathcal{T}(x_i'|x_i)} \left\{ \mathbb{E}_{q(y_i|x_i')} \log \frac{e^{U_{:y_i}^T G_{:i}/\tau}}{\sum_y e^{U_{:y}^T G_{:i}/\tau}} + H_q(y_i|x_i') \right\}$$

$$= \sum_{i=1}^{n} \mathbb{E}_{p(x_i)\mathcal{T}(x_i'|x_i)} \left\{ \mathbb{E}_{q(y_i|x_i')} \left\{ \frac{U_{:y_i}^T G_{:i}}{\tau} \right\} + H_q(y_i|x_i') \right\} + \text{const}'$$

$$= \frac{1}{\tau} \mathbb{E}_{p(x_{1:n}, x_{1:n}')} \{ Tr(QU^T G) \} + \mathbb{E}_{p(x_{1:n}, x_{1:n}')} \{ H_Q(y_{1:n}|x_{1:n}') \} + \text{const}'$$

$$= \tau \mathcal{L}_{SwAV}(Q) + \text{const}' \tag{17}$$

thus concluding the proof. $\square$

# E  Alternative View of Contrastive SSL

Let us focus the analysis on a different graphical model from the one in Section 2.1, involving an input vector $x$ and a latent embedding $z$ (cf. Figure 1). Without loss of generality, we can discard the index $i$ and focus on a single observation. We will later extend the analysis to the multi-sample case. Now, consider the conditional distribution of $x$ given $z$ expressed in the form of an energy-based model $p(x|z;\Theta) = \frac{e^{f(x,z)}}{\Gamma(z;\Theta)}$, where $f : \Omega \times \mathcal{S}^{h-1} \to \mathbb{R}$ is a score function and $\Gamma(z;\Theta) = \int_\Omega e^{f(x,z)}dx$ is the normalizing factor.[7] Based on this definition, we can obtain the following lower-bound on the data log-likelihood:

$$
\begin{aligned}
\mathbb{E}_{p(x)}\{\log p(x)\} &= KL(p(x)\|p(x;\Theta)) + \mathbb{E}_{p(x)}\{\log p(x;\Theta)\} \\
&\geq \mathbb{E}_{p(x)}\{\log p(x;\Theta)\} \\
&\geq \mathbb{E}_{p(x)q(z|x)}\{\log p(x|z;\Theta)\} - \mathbb{E}_{p(x)}\{KL(q(z|x)\|p(z;\Theta))\} \\
&= \mathbb{E}_{p(x)q(z|x)}\{\log p(x|z;\Theta)\} + \mathbb{E}_{p(x)q(z|x)}\{\log p(z;\Theta)\} \\
&= \mathbb{E}_{p(x)q(z|x)}\{f(x,z)\} + \mathbb{E}_{p(x)q(z|x)}\left\{\log \frac{p(z;\Theta)}{\Gamma(z;\Theta)}\right\} \\
&\doteq \mathrm{ELBO}_{EBM}
\end{aligned} \tag{18}
$$

where $q(z|x)$ is an auxiliary density induced by a deterministic encoding function $g : \Omega \to \mathcal{S}^{h-1}$.[8] Eq. (18) provides the basic building block to derive variational bounds on mutual information (Poole et al., 2019) as well as to obtain several popular contrastive SSL objectives.

**$\mathrm{ELBO}_{EBM}$ and variational lower bounds on mutual information**. Our analysis is similar to the one proposed in (Alemi et al., 2018), as relating $\mathrm{ELBO}_{EBM}$ to the mutual information. However, while the work in (Alemi et al., 2018) shows that the third line in Eq. (18) can be expressed as a combination of an upper and a lower bound on the mutual information and it studies its rate-distortion trade-off, our analysis considers only lower bounds to mutual information and it makes an explicit connection to them. Indeed, several contrastive objectives are based on lower bounds on the mutual information between input and latent vectors (Poole et al., 2019). We can show that $\mathrm{ELBO}_{EBM}$ is equivalent to these lower bounds under specific conditions for prior $p(z;\Theta)$ and score function $f$.

Specifically, given a uniform prior $p(z;\Theta)$ and $f(x,z) \doteq \log p(x) + \tilde{f}(x,z)$ for all admissible pair $x, z$ and for some arbitrary function $\tilde{f}$, we obtain the following equivalence (see Appendix F for the derivation):

$$
\begin{aligned}
\mathrm{ELBO}_{EBM} &= \mathbb{E}_{p(x)}\{\log p(x)\} + \mathbb{E}_{p(x)q(z|x)}\{\tilde{f}(x,z)\} \\
&\quad - \mathbb{E}_{p(x)q(z|x)}\{\log \mathbb{E}_{p(x')}\{e^{\tilde{f}(x',z)}\}\} \\
&= -H(X) + \mathcal{I}_{UBA}(X,Z)
\end{aligned} \tag{19}
$$

where $H(X) = -\mathbb{E}_{p(x)}\{\log p(x)\}$ and $\mathcal{I}_{UBA}(X,Z)$ refers to the popular Unnormalized Barber and Agakov bound on mutual information. Importantly, other well-known bounds can be derived from $\mathcal{I}_{UBA}(X,Z)$ (cf. (Poole et al., 2019) and Appendix F for further details). Consequently, maximizing $\mathrm{ELBO}_{EBM}$ with respect to the parameters of $f$ (viz. $\theta$) is equivalent to maximize a lower bound on mutual information.

**$\mathrm{ELBO}_{EBM}$ and InfoNCE (den Oord et al., 2018)**. Now, we are ready to show the derivation of the popular InfoNCE objective. By specifying a non-parametric prior[9] $p(z;\Theta) = \frac{\Gamma(z;\Theta)}{\frac{1}{n}\sum_{k=1}^n e^{f(x_k,z)}}$, where $\Gamma(\Theta) = \int \frac{\Gamma(z;\Theta)}{\frac{1}{n}\sum_{k=1}^n e^{f(x_k,z)}}dz$, we achieve the following equality (see Appendix G for the derivation):

$$
\mathrm{ELBO}_{EBM} = E_{\prod_{j=1}^n p(x_j,z_j)}\left\{\frac{1}{n}\sum_{i=1}^n \log \frac{e^{f(x_i,z_i)}}{\frac{1}{n}\sum_{k=1}^n e^{f(x_k,z_i)}}\right\} - E_{\prod_{j=1}^n p(x_j)}\{\log \Gamma(\Theta)\}
$$

---

[7]We assume that $f$ is a well-behaved function, such that the integral value $\Gamma(z;\Theta)$ is finite for all $z \in \mathcal{S}^{h-1}$.
[8]$q(z|x) = \delta(z - g(x))$.
[9]The term non-parametric refers to the fact that the samples can be regarded as parameters of the prior and therefore their number increases with the number of samples. Indeed, we have $\Theta = \{\theta; \{x_i\}_{i=1}^n\}$.

$$\doteq I_{NCE}(X,Z) - E_{\prod_{j=1}^{n} p(x_j)}\{\log \Gamma(\Theta)\} \tag{20}$$

Notably, $\text{ELBO}_{EBM}$ is equivalent to $I_{NCE}(X,Z)$ up to the term $-\log \Gamma(\Theta)$. Maximizing $\text{ELBO}_{EBM}$ has the effect to maximize $I_{NCE}(X,Z)$ and additionally to minimize $\Gamma(\Theta)$, thus ensuring that the prior $p(z;\Theta)$ self-normalizes. However, in practice, people only maximize the InfoNCE objective, disregarding the normalizing term.

## F Connection to Unnormalized Barber Agakov Bound

Firstly, we recall the derivation of the Unnormalized Barber Agakov bound (Poole et al., 2019) for the mutual information $I_{X,Z}$, adapting it to our notational convention. Secondly, we derive the equivalence relation in Eq. (19).

$$
\begin{aligned}
I_{X,Z} &= \mathbb{E}_{p(x,z)}\left\{\log \frac{p(x|z)}{p(x)}\right\} \\
&= \mathbb{E}_{p(x)q(z|x)}\left\{\log \frac{p(x|z)q(x|z)}{p(x)q(x|z)}\right\} \\
&= \mathbb{E}_{p(x)q(z|x)}\left\{\log \frac{p(x|z)}{q(x|z)}\right\} + \mathbb{E}_{p(x)q(z|x)}\left\{\log \frac{q(x|z)}{p(x)}\right\} \\
&= \mathbb{E}_{q(z)}KL\{p(x|z)\|q(x|z)\} + \mathbb{E}_{p(x)q(z|x)}\left\{\log \frac{q(x|z)}{p(x)}\right\} \\
&\geq \mathbb{E}_{p(x)q(z|x)}\left\{\log \frac{q(x|z)}{p(x)}\right\} \\
&= \mathbb{E}_{p(x)q(z|x)}\left\{\log \frac{p(x)e^{\tilde{f}(x,z)}}{p(x)Z(z)}\right\} \\
&= \mathbb{E}_{p(x)q(z|x)}\left\{\log \frac{e^{\tilde{f}(x,z)}}{Z(z)}\right\} \\
&= \mathbb{E}_{p(x)q(z|x)}\{\tilde{f}(x,z)\} - \mathbb{E}_{p(x)q(z|x)}\{\log Z(z)\} \\
&= \mathbb{E}_{p(x)q(z|x)}\{\tilde{f}(x,z)\} - \mathbb{E}_{p(x)q(z|x)}\{\log \mathbb{E}_{p(x')}\{e^{\tilde{f}(x',z)}\}\} \\
&\doteq I_{UBA}(X,Z)
\end{aligned}
$$

where we have introduced both an auxiliary encoder $q(z|x)$ and an auxiliary decoder defined as $q(x|z) = \frac{p(x)e^{\tilde{f}(x,z)}}{Z(z)}$. Now, we can use the assumptions of a uniform prior $p(z;\Theta)$ and $f(x,z) = \log p(x) + \tilde{f}(x,z)$ to achieve the following inequalities:s

$$
\begin{aligned}
\text{ELBO}_{EBM} &= \mathbb{E}_{p(x)q(z|x)}\{f(x,z)\} + \mathbb{E}_{p(x)q(z|x)}\left\{\log \frac{p(z;\Theta)}{\Gamma(z;\Theta)}\right\} \\
&= \mathbb{E}_{p(x)q(z|x)}\{f(x,z)\} - \mathbb{E}_{p(x)q(z|x)}\{\log \Gamma(z;\Theta)\} \\
&= \mathbb{E}_{p(x)q(z|x)}\{\log p(x) + \tilde{f}(x,z)\} - \mathbb{E}_{p(x)q(z|x)}\{\log \mathbb{E}_{p(x')}\{e^{\tilde{f}(x',z)}\}\} \\
&= \mathbb{E}_{p(x)q(z|x)}\{\log p(x)\} + \mathbb{E}_{p(x)q(z|x)}\{\tilde{f}(x,z)\} - \mathbb{E}_{p(x)q(z|x)}\{\log \mathbb{E}_{p(x')}\{e^{\tilde{f}(x',z)}\}\} \\
&= -H(X) + \mathbb{E}_{p(x)q(z|x)}\{\tilde{f}(x,z)\} - \mathbb{E}_{p(x)q(z|x)}\{\log \mathbb{E}_{p(x')}\{e^{\tilde{f}(x',z)}\}\} \\
&= -H(X) + I_{UBA}(X,Z)
\end{aligned}
$$

### F.1 Other Bounds

Notably, $I_{UBA}(X,Z)$ cannot be tractably computed due to the computation of $Z(z)$. Therefore, several other bounds have been derived (Poole et al., 2019) to obtain tractable estimators or optimization objectives, namely:

1. $I_{TUBA}(X, Z)$,[10] which can be used for both optimization and estimation of mutual information (obtained using the inequality $\log s \leq \eta' s - \log \eta' - 1$ for all scalar $s, \eta' > 0$)

$$
\begin{aligned}
I_{UBA}(X, Z) &= \mathbb{E}_{p(x,z)}\{\tilde{f}(x, z)\} - \mathbb{E}_{p(x)q(z|x)}\{\log Z(z)\} \\
&\geq \mathbb{E}_{p(x,z)}\{\tilde{f}(x, z)\} - \mathbb{E}_{p(x)q(z|x)}\left\{\frac{Z(z)}{\eta(z)} + \log \eta(z) - 1\right\} \\
&\doteq I_{TUBA}(X, Z)
\end{aligned}
$$

2. $I_{NWJ}(X, Z)$, which can be used for both optimization and estimation of mutual information (obtained from $I_{TUBA}(X, Z)$ by imposing $\eta(z) = e$)

$$
\begin{aligned}
I_{TUBA}(X, Z) &= \mathbb{E}_{p(x,z)}\{\tilde{f}(x, z)\} - \mathbb{E}_{p(x)q(z|x)}\left\{\frac{Z(z)}{\eta(z)} + \log \eta(z) - 1\right\} \\
&= \mathbb{E}_{p(x,z)}\{\tilde{f}(x, z)\} - \mathbb{E}_{p(x)q(z|x)}\left\{\frac{Z(z)}{e} + 1 - 1\right\} \\
&= \mathbb{E}_{p(x,z)}\{\tilde{f}(x, z)\} - \frac{1}{e}\mathbb{E}_{p(x)q(z|x)}\{Z(z)\} \\
&= \mathbb{E}_{p(x,z)}\{\tilde{f}(x, z)\} - \mathbb{E}_{p(x')p(x)q(z|x)}\{e^{\tilde{f}(x',z)-1}\} \\
&\doteq I_{NWJ}(X, Z)
\end{aligned}
$$

## G   Derivation of InfoNCE

Consider a prior $p(z) = \dfrac{\frac{1}{n}\sum_{k=1}^{n}\frac{\Gamma(z;\Theta)}{e^{f(x_k,z)}}}{\Gamma(\Theta)}$, where $\Gamma(\Theta) = \int \dfrac{\Gamma(z;\Theta)}{\frac{1}{n}\sum_{k=1}^{n}e^{f(x_k,z)}}dz$,

$$
\begin{aligned}
\text{ELBO}_{EBM} &= \mathbb{E}_{p(x)q(z|x)}\{f(x, z)\} - \mathbb{E}_{p(x)q(z|x)}\{\log \Gamma(z;\Theta)\} + \mathbb{E}_{p(x)q(z|x)}\{\log p(z;\Theta)\} \\
&= \mathbb{E}_{p(x_1)q(z|x_1)}\{f(x_1, z)\} - \mathbb{E}_{p(x_1)q(z|x_1)}\{\log \Gamma(z;\Theta)\} + \mathbb{E}_{p(x_1)q(z|x_1)}\{\log p(z;\Theta)\} \\
&= \mathbb{E}_{p(x_1)p(x_2)\cdots p(x_n)q(z|x_1)}\{f(x_1, z)\} - \mathbb{E}_{p(x_1)p(x_2)\cdots p(x_n)q(z|x_1)}\{\log \Gamma(z;\Theta)\} \\
&\quad + \mathbb{E}_{p(x_1)p(x_2)\cdots p(x_n)q(z|x_1)}\{\log p(z;\Theta)\} \\
&= \mathbb{E}_{p(x_1)p(x_2)\cdots p(x_n)q(z|x_1)}\{f(x_1, z)\} - \mathbb{E}_{p(x_1)p(x_2)\cdots p(x_n)q(z|x_1)}\left\{\log \frac{1}{n}\sum_{k=1}^{n}e^{f(x_k,z)}\right\} \\
&\quad - \mathbb{E}_{p(x_1)p(x_2)\cdots p(x_n)}\{\log \Gamma(\Theta)\} \\
&= \mathbb{E}_{p(x_1)p(x_2)\cdots p(x_n)q(z|x_1)}\{\log e^{f(x_1,z)}\} - \mathbb{E}_{p(x_1)p(x_2)\cdots p(x_n)q(z|x_1)}\left\{\log \frac{1}{n}\sum_{k=1}^{n}e^{f(x_k,z)}\right\} \\
&\quad - \mathbb{E}_{p(x_1)p(x_2)\cdots p(x_n)}\{\log \Gamma(\Theta)\} \\
&= \mathbb{E}_{p(x_1,z)p(x_2)\cdots p(x_n)}\left\{\log \frac{e^{f(x_1,z)}}{\frac{1}{n}\sum_{k=1}^{n}e^{f(x_k,z)}}\right\} \\
&\quad - \mathbb{E}_{p(x_1)p(x_2)\cdots p(x_n)}\{\log \Gamma(\Theta)\} \\
&= \mathbb{E}_{p(x_1,z_1)p(x_2,z_2)\cdots p(x_n,z_K)}\left\{\log \frac{e^{f(x_1,z_1)}}{\frac{1}{n}\sum_{k=1}^{n}e^{f(x_k,z_1)}}\right\} - \mathbb{E}_{p(x_1)p(x_2)\cdots p(x_n)}\{\log \Gamma(\Theta)\} \\
&= \frac{1}{n}\sum_{i=1}\mathbb{E}_{p(x_1,z_1)p(x_2,z_2)\cdots p(x_n,z_K)}\left\{\log \frac{e^{f(x_i,z_i)}}{\frac{1}{n}\sum_{k=1}^{n}e^{f(x_k,z_i)}}\right\} - \mathbb{E}_{p(x_1)p(x_2)\cdots p(x_n)}\{\log \Gamma(\Theta)\} \\
&= \mathbb{E}_{p(x_1,z_1)p(x_2,z_2)\cdots p(x_n,z_K)}\left\{\frac{1}{n}\sum_{i=1}\log \frac{e^{f(x_i,z_i)}}{\frac{1}{n}\sum_{k=1}^{n}e^{f(x_k,z_i)}}\right\} - \mathbb{E}_{p(x_1)p(x_2)\cdots p(x_n)}\{\log \Gamma(\Theta)\} \\
&\doteq I_{NCE}(X, Z) - \mathbb{E}_{\prod_{j=1}^{n} p(x_j)}\{\log \Gamma(\Theta)\}
\end{aligned}
$$

---

[10]T in TUBA stands for tractable.

# H    Derivation of ProtoCPC: A Lower Bound of InfoNCE

ProtoCPC (Lee, 2022) can be viewed as a lower bound of the InfoNCE objective. To save space, we use notation $\mathbb{E}$ to refer to $\mathbb{E}_{\prod_{j=1}^{n} p(x_j, z_j)}$. Therefore, we have that

$$I_{NCE} = \mathbb{E}\left\{\frac{1}{n}\sum_{i=1}^{n}\log\frac{e^{f(x_i, z_i)}}{\frac{1}{n}\sum_{k=1}^{n}e^{f(x_k, z_i)}}\right\}$$

$$= \mathbb{E}\left\{\frac{1}{n}\sum_{i=1}^{n}\log\frac{e^{\sum_{c=1}^{C}p_c^t(x_i)\log p_c^s(x_i)}}{\frac{1}{n}\sum_{k=1}^{n}e^{\sum_{c=1}^{C}p_c^t(x_k)\log p_c^s(x_i)}}\right\}$$

$$= \mathbb{E}\left\{\frac{1}{n}\sum_{i=1}^{n}\log\frac{e^{\sum_{c=1}^{C}p_c^t(x_i)\log e^{z_{c,x_i}^s/\tau_s}}\cdot e^{-\sum_{c=1}^{C}p_c^t(x_i)\log\sum_{c'=1}^{C}e^{z_{c',x_i}^s/\tau_s}}}{\frac{1}{n}\sum_{k=1}^{n}e^{\sum_{c=1}^{C}p_c^t(x_k)\log p_c^s(x_i)}}\right\}$$

$$= \mathbb{E}\left\{\frac{1}{n}\sum_{i=1}^{n}\log\frac{e^{\sum_{c=1}^{C}p_c^t(x_i)\log e^{z_{c,x_i}^s/\tau_s}}\cdot e^{-\sum_{c=1}^{C}p_c^t(x_i)\log\sum_{c'=1}^{C}e^{z_{c',x_i}^s/\tau_s}}}{\frac{1}{n}\sum_{k=1}^{n}e^{\sum_{c=1}^{C}p_c^t(x_k)\log e^{z_{c,x_i}^s/\tau_s}}\cdot e^{-\sum_{c=1}^{C}p_c^t(x_k)\log\sum_{c'=1}^{C}e^{z_{c',x_i}^s/\tau_s}}}\right\}$$

$$= \mathbb{E}\left\{\frac{1}{n}\sum_{i=1}^{n}\log\frac{e^{\sum_{c=1}^{C}p_c^t(x_i)\log e^{z_{c,x_i}^s/\tau_s}}\cdot e^{-\log\sum_{c'=1}^{C}e^{z_{c',x_i}^s/\tau_s}}}{\frac{1}{n}\sum_{k=1}^{n}e^{\sum_{c=1}^{C}p_c^t(x_k)\log e^{z_{c,x_i}^s/\tau_s}}\cdot e^{-\log\sum_{c'=1}^{C}e^{z_{c',x_i}^s/\tau_s}}}\right\}$$

$$= \mathbb{E}\left\{\frac{1}{n}\sum_{i=1}^{n}\log\frac{e^{\sum_{c=1}^{C}p_c^t(x_i)\log e^{z_{c,x_i}^s/\tau_s}}}{\frac{1}{n}\sum_{k=1}^{n}e^{\sum_{c=1}^{C}p_c^t(x_k)\log e^{z_{c,x_i}^s/\tau_s}}}\right\}$$

$$= \mathbb{E}\left\{\frac{1}{n}\sum_{i=1}^{n}\log\frac{e^{\sum_{c=1}^{C}p_c^t(x_i)z_{c,x_i}^s/\tau_s}}{\frac{1}{n}\sum_{k=1}^{n}e^{\sum_{c=1}^{C}p_c^t(x_k)z_{c,x_i}^s/\tau_s}}\right\}$$

$$\geq \mathbb{E}\left\{\frac{1}{n}\sum_{i=1}^{n}\log\frac{e^{\sum_{c=1}^{C}p_c^t(x_i)z_{c,x_i}^s/\tau_s}}{\frac{1}{n}\sum_{k=1}^{n}\sum_{c=1}^{C}p_c^t(x_k)e^{z_{c,x_i}^s/\tau_s}}\right\}$$

$$= \mathbb{E}\left\{\frac{1}{n}\sum_{i=1}^{n}\log\frac{e^{\sum_{c=1}^{C}p_c^t(x_i)z_{c,x_i}^s/\tau_s}}{\sum_{c=1}^{C}\frac{1}{n}\sum_{k=1}^{n}p_c^t(x_k)e^{z_{c,x_i}^s/\tau_s}}\right\}$$

$$= \mathbb{E}\left\{\frac{1}{n}\sum_{i=1}^{n}\log\frac{e^{\sum_{c=1}^{C}p_c^t(x_i)z_{c,x_i}^s/\tau_s}}{\sum_{c=1}^{C}q_c^t e^{z_{c,x_i}^s/\tau_s}}\right\}$$

$$= \mathbb{E}\left\{\frac{1}{n}\sum_{i=1}^{n}\log\frac{e^{p^t(x_i)^T z_{x_i}^s/\tau_s}}{\sum_{c=1}^{C}q_c^t e^{z_{c,x_i}^s/\tau_s}}\right\} \doteq I_{ProtoCPC}$$

Specifically, the second equality comes from the fact that $f(x,z) = \sum_{c=1}^{C}p_c^t(x)\log p_c^s(x)$, where $s, t$ stand for student and teacher networks, respectively, and $p_c^{\cdot}(x)$ is the $c$-th entry of the vector obtained by applying a softmax on the embedding of the corresponding network. The inequality in the derivation is obtained by applying Jensen's inequality to the denominator. Finally, $q_c^t = \frac{1}{n}\sum_{k=1}^{n}p_c^t(x_k)$ corresponds to the prototype for class $c$.

# I  Details about Negative-Free Methods

We can specify different definitions for $\mathcal{L}_{NF}(\Theta)$, namely:

1. **Barlow Twins.** The approach enforces the cross-correlation matrix to be close to the identity matrix:

$$\mathcal{L}_{NF}(\Theta) = -\|CCorr(G, G') \odot \Lambda - I\|_F^2$$

where $\|\cdot\|_F$ is the Frobenius norm, $\Lambda = J\lambda + (1-\lambda)I$, $I$ is the identity matrix, $\lambda$ is a positive hyperparameter, $J$ is a matrix of ones, $CCorr(G, G') = H^T H'$ is the sample cross-correlation matrix, $G = [g(x_1), \ldots, g(x'_n)]^T \in \mathbb{R}^{n \times h}$ and $G' = [g(x'_1), \ldots, g(x'_n)]^T \in \mathbb{R}^{n \times h}$ and

$$g(x_i) = \text{BN}(\text{Proj}(enc(x_i)))$$

Note that $BN$ is a batch normalization layer, $Proj$ is a projection head usually implemented using a multi-layer perceptron and $enc$ is the encoder.

2. **W-MSE.** This approach is similar to Barlow Twins. The main difference lie in the fact that sample cross-correlation matrix is enforced to be an identity matrix thanks to a whitening layer:

$$\mathcal{L}_{NF}(\Theta) = -Tr(G \otimes G')$$

where $Tr(G \otimes G')$ computes the trace of the outer product for matrices $G = [g(x_1), \ldots, g(x'_n)]^T \in \mathbb{R}^{n \times h}$ and $G' = [g(x'_1), \ldots, g(x'_n)]^T \in \mathbb{R}^{n \times h}$.

$$g(x_i) = \text{L2-Norm}(\text{Whitening}(\text{Proj}(enc(x_i))))$$

L2-Norm is a normalization layer based on $L_2$ norm.

3. **VICReg.** This approach attempts to simplify the architecture of Barlow Twins by introducing an invariance regularization term in the score function, thus avoiding to use a batch normalization layer:

$$\mathcal{L}_{NF}(\Theta) = -\lambda Tr((G - G') \otimes (G - G')) \\ - \mu[v(G) + v(G')] - \nu[w(G) + w(G')]$$

where $\lambda, \mu, \nu$ are positive hyperparameters and $G = [g(x_1), \ldots, g(x'_n)]^T \in \mathbb{R}^{n \times h}$ and $G' = [g(x'_1), \ldots, g(x'_n)]^T \in \mathbb{R}^{n \times h}$. The first addend enforces the representation to be invariant to the data augmentation, whereas the other two addends enforce the sample covariance matrix to be diagonal. Indeed, the second addend forces the diagonal elements of the sample covariance matrices to be unitary, namely:

$$v(G) = \sum_{j=1}^{h} \max\{0, 1 - \sqrt{Var\{G_{:j}\} + \epsilon}\}$$

where $\epsilon > 0$ is used to avoid numerical issues, $Var$ computes the variance for each column of matrix $G$. The third addend ensures that the off-diagonal elements of the sample covariance matrix approach to zero:

$$c(G) = \|Cov(G) \odot (J - I)\|_F^2$$

with $Cov(H) = G^T G$. Therefore, these last two addends have a similar behaviour to the score function of Barlow Twins. Finally, the resulting function is simplified.

$$g(x_i) = \text{Proj}(enc(x_i))$$

## J    Proof of Proposition 1

**Proposition 2.** *(Restated) Given Definition 7, the expected data log-likelihood for the probabilistic graphical model in Fig. 2(c) can be alternatively lower bounded as follows:*

$$
E_{p(x_{1:n}, x'_{1:n})}\{\log p(x_{1:n}, x'_{1:n})\} \geq -H_p(x_{1:n}) - \underbrace{\sum_{i=1}^{n} \mathbb{E}_{p(x_i)\mathcal{T}(x'_i|x_i)} \{CE(p(y_i|x'_i; \Theta), p(y_i|x_i; \Theta))\}}_{\mathcal{L}_{INV}(\Theta)}
$$

$$
\underbrace{-\sum_{i=1}^{n} CE\left(p(y_i), \frac{1}{n}\sum_{j=1}^{n} p(y_j = y_i|x_j; \Theta)\right)}_{\mathcal{L}_{PRIOR}(\Theta)} + const \tag{21}
$$

*Additionally, the corresponding maximum value for the last two addends in Eq. (21) is given by the following inequality:*[11]

$$
\mathcal{L}_{INV}(\Theta) + \mathcal{L}_{PRIOR}(\Theta) \leq -H_p(y_{1:n}) \tag{22}
$$

*Proof.* Let's start to rewrite the log-likelihood term.

$$
E_{p(x_{1:n}, x'_{1:n})}\{\log p(x_{1:n}, x'_{1:n})\} = E_{p(x_{1:n}, x'_{1:n})}\left\{\log \sum_{y_{1:n}} \prod_{i=1}^{n} p(x_i)\mathcal{T}(x'_i|x_i)p(y_i|x_i; \Theta)\right\}
$$

$$
= E_{p(x_{1:n}, x'_{1:n})}\left\{\log \prod_{i=1}^{n} \sum_{y_i} p(x_i)\mathcal{T}(x'_i|x_i)p(y_i|x_i; \Theta)\right\}
$$

$$
= -H_p(x_{1:n}) + const
$$

$$
+ \sum_{i=1}^{n} E_{p(x_i)\mathcal{T}(x'_i|x_i)}\left\{\log \sum_{y_i} p(y_i|x_i; \Theta)\right\} \tag{23}
$$

Now add the zero quantity $\log \sum_{y_{1:n}} p(y_{1:n})$ to the previous equation:

$$
\text{Eq. (23)} = -H_p(x_{1:n}) + const + \sum_{i=1}^{n} \mathbb{E}_{p(x_i)\mathcal{T}(x'_i|x_i)}\left\{\log \sum_{y_i} p(y_i|x_i; \Theta)\right\} + \log \sum_{y_{1:n}} p(y_{1:n}) \tag{24}
$$

We can lower bound the previous equation by exploiting the fact that $\sum_z p(z) \geq \sum_z p(z)q(z)$ for any given auxiliary discrete distribution $q$, viz.:

$$
\text{Eq. (24)} \geq -H_p(x_{1:n}) + const + \sum_{i=1}^{n} \mathbb{E}_{p(x_i)\mathcal{T}(x'_i|x_i)}\left\{\log \sum_{y_i} q(y_i|x'_i)p(y_i|x_i, \Theta)\right\}
$$

$$
+ \log \sum_{y_{1:n}} p(y_{1:n})q(y_{1:n}) \tag{25}
$$

Now, by applying Jensen's inequality to the last two addends in Eq. (25), by defining $q(y_{1:n}) = \frac{1}{n}\sum_{j=1}^{n} p(y_j|x_j, \Theta)$ and considering Definition 10, we obtain the following lower bound:

$$
\text{Eq. (24)} \geq -H_p(x_{1:n}) + const + \sum_{i=1}^{n} \mathbb{E}_{p(x_i)\mathcal{T}(x'_i|x_i)}\left\{\sum_{y_i} p(y_i|x'_i, \Theta)\log p(y_i|x_i, \Theta)\right\}
$$

$$
+ \sum_{y_{1:n}} p(y_{1:n})\log \frac{1}{n}\sum_{j=1}^{n} p(y_j|x_j, \Theta) \tag{26}
$$

---

[11]Here, we assume that the predictive model $p(y|x; \Theta)$ has enough capacity to achieve the optimal solution.

Additionally, by substituting $p(y_{1:n}) = \prod_{i=1}^{n} p(y_i)$ into the above bound, we achieve the following equality:

$$\text{Eq. (26)} = -H_p(x_{1:n}) + \text{const} + \sum_{i=1}^{n} \mathbb{E}_{p(x_i)\mathcal{T}(x'_i|x_i)} \left\{ \sum_{y_i} p(y_i|x'_i; \Theta) \log p(y_i|x_i; \Theta) \right\}$$

$$+ \sum_{i=1}^{n} \sum_{y_i} p(y_i) \log \left( \frac{1}{n} \sum_{j=1}^{n} p(y_j = y_i|x_j; \Theta) \right) \tag{27}$$

And by rewriting the last two addends in Eq. (27) using the definition of cross-entropy, we obtain our final result.

Now, we can conclude the proof by looking at the maxima for $\mathcal{L}_{INV}$ and $\mathcal{L}_{PRIOR}$. Indeed, we observe that both terms compute a negative cross-entropy between two distributions. By leveraging the fact that $CE(p, q) = H_p + KL(p\|q)$ for arbitrary distributions $p, q$, we can easily see that the maximum of $\mathcal{L}_{INV}$ is attained when the term is 0 (corresponding to minimal entropy and minimal KL), whereas the maximum of $\mathcal{L}_{PRIOR}$ is attained when the term is equal to $-H_p(y_i)$ (corresponding to minimal KL). $\qquad\square$

## K Proof of Theorem 1

*Proof.* The overall strategy to prove the statements relies on the evaluation of the loss terms over the three failure modes and on checking whether these attain their corresponding maxima.

Let's start by proving statement a and recalling that $\mathcal{L}_{GEN}(\Theta, \mathcal{D}) = -CE(p, p_\Theta)$. Firstly, we test for failure mode 1 (i.e. representational collapse). We observe that for all $x \in \mathbb{R}^d$

$$p_\Theta(x) = \frac{\sum_{y=1}^{c} e^{U_{:y}^T g(k)/\tau}}{\Gamma(\Theta)}$$

thus $p_\Theta(x)$ assigns constant mass everywhere. Clearly, $p_\Theta$ is different from $p$. Therefore, $-CE(p, p_\Theta) < -CE(p, p)$ and failure mode 1 is not admissible. Secondly, we test for failure mode 2 (i.e. cluster collapse). We can equivalently rewrite the definition of cluster collapse by stating that there exists $j \in \{1, \ldots, c\}$, such that for all $x \in \mathbb{R}^d$ and for all $y \neq j$, $U_{:j}^T g(x) - U_{:y}^T g(x) \to \infty$. Additionally, we observe that

$$p_\Theta(x) = \frac{\sum_{y=1}^{c} e^{U_{:y}^T g(x)/\tau}}{\int \sum_{y=1}^{c} e^{U_{:y}^T g(x)/\tau} dx}$$

$$= \frac{e^{U_{:j}^T g(x)/\tau} \left[ 1 + \sum_{y \neq j} \cancelto{0}{e^{(U_{:y}^T g(x) - U_{:j}^T g(x))/\tau}} \right]}{\int e^{f_j(\xi_x)/\tau} \left[ 1 + \sum_{y \neq j} \cancelto{0}{e^{(U_{:y}^T g(x) - U_{:j}^T g(x))/\tau}} \right] dx}$$

$$= \frac{e^{U_{:j}^T g(x)/\tau}}{\int e^{U_{:j}^T g(x)/\tau} dx} \tag{28}$$

where we have used the failure mode condition to obtain the last equality. Now, note that Eq. (28) defines a standard energy-based model. Consequently, given enough capacity for the predictive model, it is trivial to verify that there exists $\Theta$ such that the condition about failure mode is met and $p_\Theta$ is equal to $p$. Cluster collapse is therefore an admissible solution. Thirdly, we test for the inconsistency of cluster assignments. Indeed, we have that

$$\mathcal{L}_{GEN}(\Theta, \mathcal{D}) = \sum_{i=1}^{n} \mathbb{E}_{p(x_i)} \left\{ \log p_\Theta(x_i) \right\}$$

$$= \sum_{i=1}^{n} \mathbb{E}_{p(x_i)} \left\{ \log \frac{\sum_{y=1}^{c} e^{t_i(y)/\tau}}{\int \sum_{y=1}^{c} e^{t_i(y)/\tau} dx} \right\}$$

$$= \sum_{i=1}^{n} \mathbb{E}_{p(x_{1:n})} \left\{ \log \frac{\sum_{y=1}^{c} e^{t_i(y)/\tau}}{\int \sum_{y=1}^{c} e^{t_i(y)/\tau} dx} \right\} \tag{29}$$

where $t_i(y) = U_{:y}^T g(x_i)$. Similarly, we have that

$$\mathcal{L}_{GEN}(\Theta, \mathcal{D}^\pi)$$

$$\underbrace{=}_{\text{from Eq. (29)}} \sum_{i=1}^{n} \mathbb{E}_{p(x_{1:n})} \left\{ \log \frac{\sum_{y=1}^{c} e^{t_{\pi(i)}(y)/\tau}}{\int \sum_{y=1}^{c} e^{t_{\pi(i)}(y)/\tau} dx} \right\}$$

$$= \sum_{i=1}^{n} \mathbb{E}_{p(x_{\pi(i)})} \left\{ \log \frac{\sum_{y=1}^{c} e^{t_{\pi(i)}(y)/\tau}}{\int \sum_{y=1}^{c} e^{t_{\pi(i)}(y)/\tau} dx} \right\}$$

$$\underbrace{=}_{l \doteq \pi(i)} \sum_{l=1}^{n} \mathbb{E}_{p(x_l)} \left\{ \log \frac{\sum_{y=1}^{c} e^{t_l(y)/\tau}}{\int \sum_{y=1}^{c} e^{t_l(y)/\tau} dx} \right\}$$

$$= \mathcal{L}_{GEN}(\Theta, \mathcal{D})$$

Hence, failure mode 3 is an admissible solution.

Let's continue by proving statement b and recalling that

$$\mathcal{L}_{INV}(\Theta, \mathcal{D}) = - \sum_{i=1}^{n} \mathbb{E}_{p(x_i)\mathcal{T}(x_i'|x_i)} \left\{ CE(p(y_i|x_i';\Theta), p(y_i|x_i;\Theta)) \right\} \tag{30}$$

Firstly, we test for representational collapse. In this case, we have that for all $i \in \{1, \ldots, n\}$

$$p(y_i|x_i;\Theta) = p(y_i|x_i';\Theta) = \text{Softmax}(U^T k/\tau)$$

Based on this result, we observe that the cross-entropy terms in Eq. (30) can be made 0 by proper choice of $k$. Therefore, representational collapse is an admissible solution. Secondly, we test for cluster collapse. Here, it is easy to see that the cross-entropy terms in Eq. (30) are all 0. Therefore, also cluster collapse is admissible. Thirdly, we test for the inconsistency of cluster assignments. On one hand, we have that the cross-entropy terms for $\mathcal{L}_{INV}(\Theta, \mathcal{D}))$ in Eq. (30) can be rewritten in the following way:

$$CE(p(y_i|x_i';\Theta), p(y_i|x_i;\Theta))$$

$$= CE \left( \frac{e^{t_i'(y_i)/\tau}}{\sum_{y=1}^{c} e^{t_i'(y)/\tau}}, \frac{e^{t_i(y_i)/\tau}}{\sum_{y=1}^{c} e^{t_i(y)/\tau}} \right) \tag{31}$$

and the optimal solution is achieved only when $t_i' = t_i$ for all $i \in \{1, \ldots, n\}$. On the other hand, the cross-entropy terms for $\mathcal{L}_{INV}(\Theta, \mathcal{D}^\pi)$ are given by the following equality:

$$CE(p(y_i|x_i';\Theta), p(y_i|x_i;\Theta))$$

$$= CE \left( \frac{e^{t_i'(y_i)/\tau}}{\sum_{y=1}^{c} e^{t_i'(y)/\tau}}, \frac{e^{t_{\pi(i)}(y_i)/\tau}}{\sum_{y=1}^{c} e^{t_{\pi(i)}(y)/\tau}} \right) \tag{32}$$

However, the optimal solution cannot be achieved in general as $t_i' \neq t_{\pi(i)}$ for some $i \in \{1, \ldots, n\}$.[12] Therefore, $\mathcal{L}_{INV}$ is not permutation invariant to cluster assignments.

Let's conclude by proving statement c and recalling that

$$\mathcal{L}_{PRIOR}(\Theta, \mathcal{D}) = - \sum_{i=1}^{n} CE \left( p(y_i), \frac{1}{n} \sum_{l=1}^{n} p(y_l = y_i|x_l;\Theta) \right) \tag{33}$$

---

[12]Indeed, note that $t_i' = t_{\pi(i)}$ for all $i$ occurs only when we are in one of the first two failure modes.

Firstly, we test for representational collapse. One can easily observe that if $enc(x) = k$ for all $x \in \mathbb{R}^d$, $p(y|x; \Theta)$ becomes uniform, namely $p(y|x; \Theta) = 1/c$ for all $y \in \{1, \ldots, c\}$. Consequently, $\frac{1}{n} \sum_{l=1}^{n} p(y_l = y_i | x_l; \Theta) = 1/c$ for all $i \in \{1, \ldots, n\}$. Now, since $p(y_i) = 1/c$ for all $i \in \{1, \ldots, n\}$, the cross-entropy terms in Eq. (33) reach their maximum value $-H_p(y_i)$ for all $i \in \{1, \ldots, n\}$. Therefore, representational collapse attains the global maximum of $\mathcal{L}_{PRIOR}$ and is an admissible solution. Secondly, we test for cluster collapse. By using the definition of cluster collapse, we observe that

$$\frac{1}{n} \sum_{l=1}^{n} p(y_l = y_i | x_l; \Theta) = \left\{ \begin{array}{ll} 0 & y_i = j \\ 1 & y_i \neq j \end{array} \right. \tag{34}$$

Therefore, the resulting distribution is non-uniform, differently from $p(y_i)$. The cross-entropy terms in Eq. (33) are not optimized and cluster collapse is not admissible. Thirdly, we test for the inconsistency of cluster assignments. We observe that

$$\frac{1}{n} \sum_{l=1}^{n} p(y_l = y_i | x_l; \Theta) = \frac{1}{n} \sum_{l=1}^{n} \frac{e^{t_l(y_i)/\tau}}{\sum_{y=1}^{c} e^{t_l(y)/\tau}}$$
$$= \frac{1}{n} \sum_{l=1}^{n} \frac{e^{t_{\pi(l)}(y_i)/\tau}}{\sum_{y=1}^{c} e^{t_{\pi(l)}(y)/\tau}} \tag{35}$$

which is permutation invariant to cluster assignments. Consequently, also $\mathcal{L}_{PRIOR}(\Theta, \mathcal{D}) = \mathcal{L}_{PRIOR}(\Theta, \mathcal{D}^{\pi})$. This concludes the proof. □

## L   Hyperparameters for Synthetic Data

For the backbone *enc*, we use a MLP with two hidden layers and 100 neurons per layer, an output layer with 2 neurons and ReLU activation functions. For the projection head *proj* (*f* for GEDI and its variants), we use a MLP with one hidden layer and 4 neurons and an output layer with 2 neurons (batch normalization is used in all layers for Barlow and SwAV as required by their original formulation). All methods use a batch size of 400. Baseline JEM (following the original paper): Number of iterations $20K$; learning rate $1e-3$; Adam optimizer($\beta_1 = 0.9$, $\beta_2 = 0.999$); SGLD steps 10; buffer size 10000; reinitialization frequency 0.05; SGLD step-size $\frac{0.01^2}{2}$; SGLD noise 0.01. For the self-supervised learning methods, please refer to Table 7. We also provide an analysis of sensitivity to hyperparameters for GEDI (Figure 8).

Table 7: Hyperparameters used in the synthetic experiments.

| Methods | Barlow | SwAV | GEDI *no gen* | GEDI |
|---|---|---|---|---|
| Iters | | | $20k$ | |
| Learning rate | | | $1e-3$ | |
| Optimizer | | Adam $\beta_1 = 0.9, \beta_2 = 0.999$ | | |
| Data augmentation noise $\sigma$ | | | 0.03 | |
| SGLD steps $T$ | - | - | 1 | 1 |
| Buffer size $|B|$ | - | - | 10000 | 10000 |
| Reinitialization frequenc | - | - | 0.05 | 0.05 |
| SGLD step size | - | - | $\frac{0.01^2}{2}$ $\frac{0.01^2}{2}$ | $\frac{0.01^2}{2}$ |
| SGLD noise | - | - | 0.01 | 0.01 |
| Weight for $\mathcal{L}_{GEN}$ | - | - | 1 | 1 |
| Weight for $\mathcal{L}_{INV}$ | - | - | 50 | 50 |
| Weight for $\mathcal{L}_{PRIOR}$ | - | - | 10 | 10 |

## M   Hyperparameters for SVHN, CIFAR10, CIFAR100

For the backbone *enc*, we use a ResNet with 8 layers as in (Duvenaud et al., 2021), where its architecture is shown in Table 8. For the projection head *proj* (*f* for GEDI and its variants), we use a MLP with one hidden

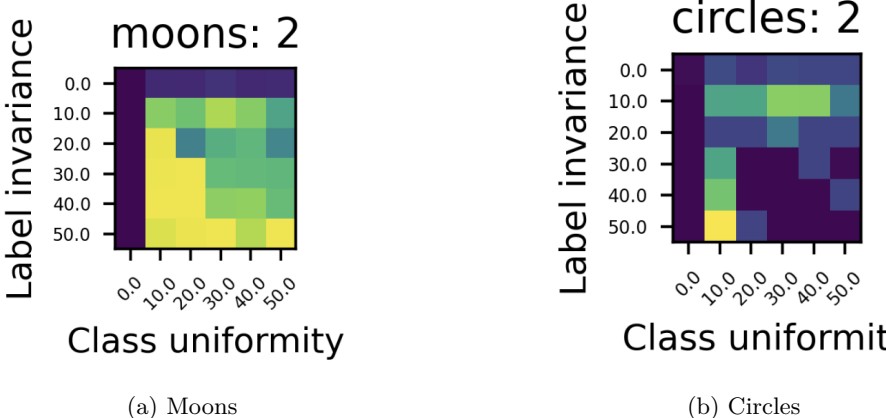

(a) Moons

(b) Circles

Figure 8: Sensitivity analysis on the discriminative performance of GEDI fr different loss weights (in the range $\{0, 10, 20, 30, 40, 50\}$). Performance are averaged over 5 different random seeds. Yellow means perfect NMI.

Table 8: Resnet architecture. Conv2D(A,B,C) applies a 2d convolution to input with B channels and produces an output with C channels using stride (1, 1), padding (1, 1) and kernel size (A, A).

| Name | Layer | Res. Layer |
|---|---|---|
| Block 1 | Conv2D(3,3,F) LeakyRELU(0.2) Conv2D(3,F,F) AvgPool2D(2) | AvgPool2D(2) Conv2D(1,3,F) no padding |
| | Sum | |
| Block 2 | LeakyRELU(0.2) Conv2D(3,F,F) LeakyRELU(0.2) Conv2D(3,F,F) AvgPool2D(2) | |
| Block 3 | LeakyRELU(0.2) Conv2D(3,F,F) LeakyRELU(0.2) Conv2D(3,F,F) | |
| Block 4 | LeakyRELU(0.2) Conv2D(3,F,F) LeakyRELU(0.2) Conv2D(3,F,F) AvgPool2D(all) | |

layer and $2 * F$ neurons and an output layer with $F$ neurons (batch normalization is used in all layers for Barlow and SwAV as required by their original formulation + final $L_2$ normalization). $F = 128$ for SVHN, CIFAR-10 (1 million parameters) and $F = 256$ for CIFAR-100 (4.1 million parameters). For JEM, we use the same settings of (Duvenaud et al., 2021). All methods use a batch size of 64. Baseline JEM (following the original paper): number of epochs 20, 200, 200 for SVHN, CIFAR-10, CIFAR-100, respectively; learning rate $1e - 4$; Adam optimizer; SGLD steps 20; buffer size 10000; reinitialization frequency 0.05 ; SGLD step-size

Table 9: Hyperparameters (in terms of sampling, optimizer, objective and data augmentation) used in all experiments.

| Class | Name param. | SVHN | CIFAR-10 | CIFAR-100 | MNIST | Addition |
|---|---|---|---|---|---|---|
| Data augment. | Color jitter prob. | 0.1 | 0.1 | 0.1 | 0.1 | 0.1 |
| | Gray scale prob. | 0.1 | 0.1 | 0.1 | 0.1 | 0.1 |
| | Random crop | Yes | Yes | Yes | Yes | Yes |
| | Additive Gauss. noise (std) | 0.03 | 0.03 | 0.03 | 0.2 | 0.2 |
| | Random horizontal flip | No | Yes | Yes | No | No |
| SGLD | SGLD iters | 20 | 20 | 20 | 20 | 20 |
| | Buffer size | 10k | 10k | 10k | 10k | 10k |
| | Reinit. frequency | 0.05 | 0.05 | 0.05 | 0.05 | 0.05 |
| | SGLD step-size | 1 | 1 | 1 | 1 | 1 |
| | SGLD noise | 0.01 | 0.01 | 0.01 | 0.01 | 0.01 |
| Optimizer | Batch size | 64 | 64 | 64 | 60 | 60 |
| | Epochs | 20 | 200 | 200 | * | * |
| | Adam $\beta_1$ | 0.9 | 0.9 | 0.9 | 0.9 | 0.9 |
| | Adam $\beta_2$ | 0.999 | 0.999 | 0.999 | 0.999 | 0.999 |
| | Learning rate | $1e-4$ | $1e-4$ | $1e-4$ | $1e-4$ | $1e-4$ |
| | L2 regularization | 0 | 0 | 0 | $1e-4$ | $1e-4$ |
| Weights for losses | $\mathcal{L}_{GEN}$ | 1 | 1 | 1 | 1 | 1 |
| | $\mathcal{L}_{INV}$ | 50 | 50 | 50 | 50 | 0 |
| | $\mathcal{L}_{PRIOR}$ | 25 | 25 | 50 | 25 | 400 |
| | $\mathcal{L}_{NESY}$ | - | - | - | 0 | 25 |

*: For the 3 different dataset sizes (100, 1000, 10000), we trained for 500, 100 and 25 epochs respectively.

1; SGLD noise 0.01; data augmentation (Gaussian noise) 0.03. For the self-supervised learning methods, please refer to Table 9.

# N    Additional Experiments on SVHN, CIFAR-10, CIFAR-100

We conduct a linear probe evaluation of the representations learnt by the different models Table 10. These experiments provide insights on the capabilities of learning representations producing linearly separable classes. From Table 10, we observe a large difference in results between Barlow and SwAV. Our approach provides interpolating results between the two approaches.

We also provide additional qualitative analyisis on the generation performance on SVHN and CIFAR-100. Please, refer to Figure 9 and Figure 10.

Finally, we evaluate the performance in terms of OOD detection, by following the same methodology used in (Grathwohl et al., 2020). We use the OOD score criterion proposed in (Grathwohl et al., 2020), namely $s(x) = -\|\frac{\partial \log p_\Psi(x)}{\partial x}\|_2$. From Table 5, we observe that GEDI achieves almost optimal performance. While these results are exciting, it is important to mention that they are not generally valid. Indeed, when training on CIFAR-10 and performing OOD evaluation on the other datasets, we observe that all approaches achieve similar performance both on CIFAR-100 and SVHN, suggesting that all datasets are considered in-distribution, see Table 11. A similar observation is obtained when training on CIFAR-100 and evaluating on CIFAR-10 and SVHN, see Table 12. Importantly, this is a phenomenon which has been only recently observed by the scientific community on generative models. Tackling this problem is currently out of the scope of this work. For further discussion about the issue, we point the reader to the works in (Nalisnick et al., 2019).

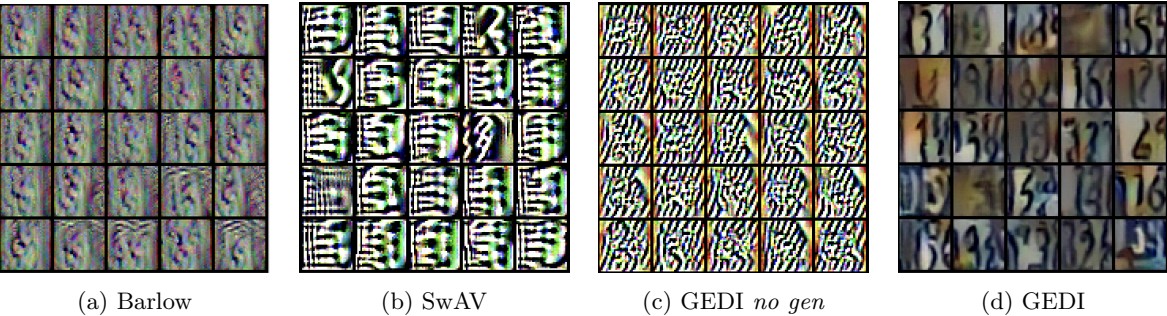

| (a) Barlow | (b) SwAV | (c) GEDI *no gen* | (d) GEDI |

Figure 9: Qualitative visualization of the generative performance for the different discriminative strategies on SVHN. Results are obtained by running Stochastic Langevin Dynamics for 500 iterations.

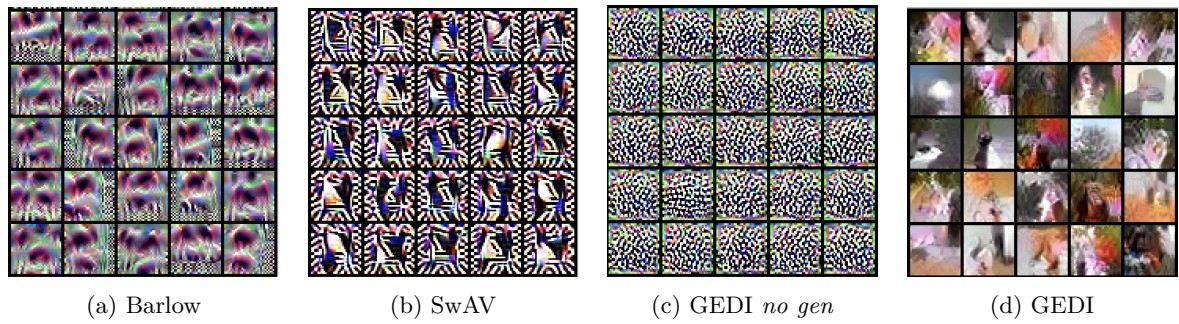

| (a) Barlow | (b) SwAV | (c) GEDI *no gen* | (d) GEDI |

Figure 10: Qualitative visualization of the generative performance for the different discriminative strategies on CIFAR-100. Results are obtained by running Stochastic Langevin Dynamics for 500 iterations.

Table 10: Supervised linear evaluation in terms of accuracy on test set (SVHN, CIFAR-10, CIFAR-100). The linear classifier is trained for 100 epochs using SGD with momentum, learning rate $1e-3$ and batch size 100. Mean and standard deviations are computed over results from 5 different initialization seeds.

| Dataset | JEM | Barlow | SwAV | GEDI *no gen* | GEDI |
|---------|-----|--------|------|---------------|------|
| SVHN | 0.20±0.00 | **0.75±0.01** | 0.44±0.04 | 0.59±0.02 | 0.54±0.01 |
| CIFAR-10 | 0.24±0.00 | **0.65±0.00** | 0.50±0.02 | 0.63±0.01 | 0.63±0.01 |
| CIFAR-100 | 0.03±0.00 | 0.27±0.01 | 0.14±0.01 | **0.31±0.01** | **0.31±0.01** |

Table 11: OOD detection in terms of AUROC on test set (SVHN, CIFAR-100). Training is performed on CIFAR-10.

| Dataset | JEM | Barlow | SwAV | GEDI *no gen* | GEDI |
|---------|-----|--------|------|---------------|------|
| SVHN | 0.44 | 0.32 | **0.62** | 0.11 | 0.57 |
| CIFAR-100 | 0.53 | 0.56 | 0.51 | 0.51 | **0.61** |

Table 12: OOD detection in terms of AUROC on test set (SVHN, CIFAR-10). Training is performed on CIFAR-100.

| Dataset | JEM | Barlow | SwAV | GEDI *no gen* | GEDI |
|---------|-----|--------|------|---------------|------|
| SVHN | 0.44 | 0.45 | 0.30 | **0.55** | 0.53 |
| CIFAR-10 | **0.49** | 0.43 | 0.47 | 0.46 | 0.48 |

## O  Details on the MNIST addition experiment.

The hyperparameters for the experiment without and with the NeSy constraint are as reported in Table 9. The data was generated by uniformly sampling pairs $a, b$ such that $0 \leq a \leq 9$, $0 \leq b \leq 9$ and $0 \leq$

$a + b \leq 9$. For each triplet $(a, b, c)$, we assigned to $a, b, c$, random MNIST images with corresponding labels, without replacement. For the experiment with the NeSy constraint, we use a slightly different uniformity regularization. We maximize the entropy of the mean output distribution for each batch, cf. Manhaeve et al. (2021).

### O.1 Replacing $\mathcal{L}_{INV}$ with $\mathcal{L}_{NESY}$

For these experiments, we replace $\mathcal{L}_{INV}$ with $\mathcal{L}_{NESY}$ as the NeSy loss also forces a clustering of the digits, making $\mathcal{L}_{INV}$ redundant. Figure 11 shows that both the NeSy Loss and the clustering loss progress similarly.

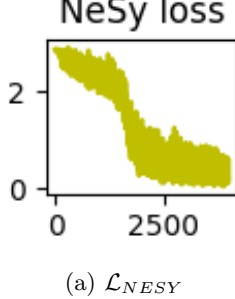 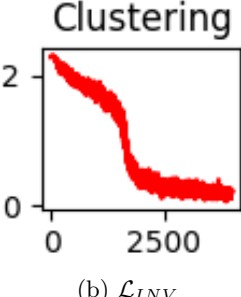

(a) $\mathcal{L}_{NESY}$        (b) $\mathcal{L}_{INV}$

Figure 11: Comparing $\mathcal{L}_{NESY}$ with $\mathcal{L}_{INV}$ on the Addition experiment, where only the NeSy loss is optimized.

