# OpenReview forum: "Unifying Self-Supervised Clustering and Energy-Based Models"
_TMLR — Accepted by TMLR_

### Review · Reviewer_mBoX · 2025-04-23

**Summary Of Contributions:**

This paper attempts to bridge the gap between SSL models and generative models in a unified framework, and demonstrate an instantiation with SSL clustering and energy-based models. The proposed method, GEDI, is tested on two toy data distributions and three simple yet classic computer vision datasets. The results were evaluated with normalized mutual information (NMI) and Frechet Inception Distance (FID) and demonstrate that the model can generate and also perform the discriminative downstream task.

**Audience:**

Yes

**Broader Impact Concerns:**

The reviewer sees no immediate broader impact concerns.

**Claims And Evidence:**

Yes

**Requested Changes:**

Discussion on the missed key reference.

In the experiments, include baseline(s) that combine SSL and generative modeling if possible.

Add classification accuracy as part of the SSL evaluation.

**Strengths And Weaknesses:**

Strengths:

1. Combining SSL and generative models is a problem of significant importance, and this paper builds connections from theoretical perspectives.

2. Experiments on toy data and simple yet classic computer vision datasets demonstrate that GEDI works as expected.

Weakness:

1. Missing related work. The paper mainly discusses related work in SSL, but little on methods that combine SSL and generative models. A missed key reference is [1].

2. Missing baselines. Similar to W.1, the authors only compare to SSL baselines, but no other methods that combine SSL and generative models. Additionally, classification accuracy (KNN, linear probing) is conceived standard in benchmarking SSL methods. Using only NMI is valid, but it will be better if the accuracies can also be presented.

3. Writing conciseness. It is well-known that many versions of SSL objectives are similar but have subtle differences. So the reviewer doubts the necessity of re-introducing many definitions for different kinds of SSL models. Since the paper focuses specifically on SSL clustering and energy-based models, it might be sufficient and more focused if the discussion is kept to the key message and key contribution of the paper.


[1] Tong, S., Dai, X., Chen, Y., Li, M., Li, Z., Yi, B., LeCun, Y. and Ma, Y., 2022. Unsupervised learning of structured representations via closed-loop transcription. arXiv preprint arXiv:2210.16782.

---

> ### Author Response · Authors · 2025-05-30
> **Answer to review**
>
> Thank you for taking the time to review our work and for your thoughtful feedback. We have submitted a revised version of our manuscript, in which all modifications to the main text are highlighted in purple. Please find below our detailed responses to the requested changes.
>
> **Reference** Thank you for pointing out the additional reference. The paper in [1] investigates how to train an autoencoder to solve both discriminative and generative tasks by proposing an objective function defined entirely in the latent space. Instead of combining a discriminative encoder with a generative decoder, our approach trains a single classifier network to operate in both a discriminative and generative manner. To the best of our knowledge, our work is the first to achieve such dual functionality, marking a novel attempt at unifying self-supervised learning with energy-based models. Crucially, the advantage of this unification lies not only in enabling both discriminative and generative capabilities, but also in providing robustness to out-of-distribution detection, an area where energy-based models particularly excel. This robustness in concept identification is essential for tackling the symbol grounding problem in neuro-symbolic tasks, as demonstrated in Section 6.4 of the main paper. Thank you again for the reference, we have cited the work and updated our related work accordingly.
>
> **Reference** \
> [1] Tong, S., Dai, X., Chen, Y., Li, M., Li, Z., Yi, B., LeCun, Y. and Ma, Y., 2024. Unsupervised learning of structured representations via closed-loop transcription. First Conference on Parsimony and Learning 2024.
>
> **Classification accuracy** We have reported the supervised linear probing performance for all methods in Table 10 of Appendix N. Our approach is competitive with the best-performing baseline, Barlow Twins, which belongs to the family of feature decorrelation methods (referred to as negative-free SSL in our paper). Notably, our method outperforms the clustering-based competitor, SwAV.
>
> We hope we have addressed your concerns effectively. Please let us know if there is anything you would like to discuss further.

---

### Review · Reviewer_eh7p · 2025-04-25

**Summary Of Contributions:**

A framework combining energy-based generative modeling and clustering-based representation learning is proposed. It is based on a probabilistic formulation of cluster-based self-supervised learning. The integration with generative modeling is introduced through lower-bounding the entropy term appearing in the probabilistic formulation. The authors analyze the derived objective function and show it can avoid three failure modes that cluster-based SSL often encounters. The utility of the proposed method is demonstrated with empirical evidence.

**Audience:**

Yes

**Claims And Evidence:**

Yes

**Requested Changes:**

Some parts of the paper seem to overly rely on a reader's knowledge of recent SSL studies. Adding more basic explanation would be highly helpful.

1) In Section 2, the ground truth distributions and the model distributions of the three SSL objectives are introduced without much explanation. Adding a brief description of each SSL objective in the main text, e.g., on the idea behind each approach, would be helpful.

2) In Section 2, the definitions of $\Theta$ and $\theta$ are unclear. Such unclearness may be inevitable to maintain the generality, but more intuition on what it is for each case would be useful.

3) The proof of Theorem 1, at least a sketch of it, should be included in the main text, as it is one of the main claims of the paper.

Below are minor things.

- Please define the notation $\{ x_i \}_i^n$.
- Every now and then the reference numbers are wrong. E.g., "Eq. (21)" in Page 7, "Definition 12" (and 13, 14) in Page 9, ... Maybe there are others too.

**Strengths And Weaknesses:**

### Strengths

- Unification of different approaches to SSL is interesting and important, and the proposed probabilistic formulation looks reasonable for that purpose.
- The experiment is done not only with synthetic data but also with more realistic image data. Also the method is applied to a more non-trivial task.
- Related work is intensively surveyed.

### Weaknesses

- Although the paper is nicely written, as a reader who is not very familiar with SSL literature, I felt adding a little more explanation would be helpful, as detailed in the next form.
- The first two views presented in Section 2 (contrastive and negative-free SSL) have much less attention than cluster-based SSL, in the subsequent parts of the paper. It is not problematic to focus on one of them, but it would further strenghten the paper to give some more discussion on the other SSL approaches. For example, can we think of the unification of contrastive (or negative-free) SSL with generative modeling as well? What is the potential difficulty? Any existing work on it? etc.

---

> ### Author Response · Authors · 2025-05-30
> **Answer to review**
>
> Thank you for taking the time to review and understand our work, and for providing valuable suggestions aimed at improving the clarity of the presentation. We have submitted a revised version of our manuscript in which all modifications to the main text are highlighted in purple. Please find below detailed comments to the requested changes.
>
> **Section 2** Thank you for your suggestions. We have added a brief explanatory sentence for each class of SSL approaches to provide intuition behind their definitions and objectives. Additionally, we have explicitly defined $\Theta$ and $\theta$ for each class as follows:
> - For contrastive SSL, $\Theta =(\theta,(x_i))$, where $\theta$ denotes the parameters of the function $g$. Definition 2 and Lemma 1 have been revised accordingly.
> - For negative-free SSL, $\Theta =(\theta)$, where $\theta$ represents the parameters of the backbone encoder $enc$ and the composed function $g$ (backbone plus projector). We have updated Definition 4 and Lemma 2 to clarify this.
> - For cluster-based SSL, $\Theta = (\theta, U)$. Definition 5 has been modified to more clearly express the dependence of $g$ on $\theta$.
>
> **Theorem 1 proof** Thank you. We have added a proof sketch in the main paper. The core idea is to evaluate the loss terms at the solutions associated with each failure mode, and to verify whether these solutions correspond to global minima of the respective loss functions.
>
> **Typos** Thank you for pointing these out. We have corrected the references to equations and definitions in the main text, and have thoroughly proofread the appendices as well.
>
> We hope we have addressed your concerns effectively. Please let us know if there is anything you would like to discuss further.

---

### Review · Reviewer_6oNa · 2025-05-16

**Summary Of Contributions:**

This paper studies probabilistic interpretations of self-supervised learning methods and proposes a novel self-supervised learning paradigm based on this analysis. The paper begins with discussing three families of self-supervised algorithms (Contrastive, Negative-Free, and Cluster-Based) and presents a probabilistic interpretation of each family, which incorporate a wide variety of popular SSL algorithms as special cases. Furthermore, the work observes that the entropy term that is often bypassed in SSL frameworks can be estimated by a loss term of involving an unconditional EBM derived from the probabilistic SSL interpretation. Next, the GEDI method is proposed, which reformulates the Cluster-Based learning objective into three terms: a term that encourages two views of the same image to be sorted to the same cluster, a term that encourages marginal cluster labels to follow a uniform prior, and an EBM term that gives the cross entropy between the marginal model and marginal data distributions. The first two terms are learned in a supervised way, and the last term uses standard MCMC EBM learning methods. An analysis of failure modes presents three forms: representational collapse, cluster collapse, and label inconsistency. The failure modes are prevented by the generative, prior, and cluster classification terms respectively. The experiment section first presents results on toy experiments, ablating the three loss terms and outperforming related clustering methods. The next experiment examines SSL on SVHN, CIFAR-10, and CIFAR-100, showing better clustering performance than related SSL methods and better generation results (of which the presented method is the only one that can resemble the training data). OOD detection is also examined. Finally, a symbol grounding experiment is performed on MNIST, showing that incorporating GEDI with existing neural grounding frameworks shows significant improvement over the baseline.

**Audience:**

Yes

**Broader Impact Concerns:**

None were presented, but this does not impact my assessment of the paper.

**Claims And Evidence:**

Yes

**Requested Changes:**

My suggested changes would be to strengthen the experimental section by including large scale data like ImageNet (or at least smaller version like Tiny Imagenet) and to investigate linear probing, few-shot, and transfer learning with the SSL features.

**Strengths And Weaknesses:**

**Strengths**

1. The presentation of SSL methods as probabilistic models provides a nice unifying view of a wide variety of methods. This interpretation seems natural and could lead to developing new methods, like GEDI.
2. The GEDI loss formulation seems more natural than the cluster-based family presented in Section 2.3. Each loss term has a clear and natural interpretation, and the formulation avoids ad-hoc fixes like stop gradients and alternating optimizations used in prior work.
3. The analysis of the failure modes is quite nice and very intuitive. It is clear how each loss term has a role in preventing a kind of learning collapse that cannot be avoided by the other loss terms.
4. The experiment section shows convincing ablation of loss terms and comparison to related cluster-based SSL. On the scale of CIFAr-10 and CIFAR-100, the proposed method can outperform other clustering SSL by a significant in some cases and it is the only method with any kind of generative capabilities (however, the generative capability is very limited, and I am not sure how essential it is). The neuro-symbolic grounding experiment offers some evidence that SSL can improve existing methods in this area.

**Weaknesses**

The main weaknesses are related to the experimental evaluation.

1. The scale of the proposed method is limited to small-scale datasets like CIFAR-10 and CIFAR-100. ImageNet scale experiments are standard in this area. How well does the method perform at scale? Does the EBM loss term and MCMC learning limit the scalability of the proposed method?
2. Standard SSL experiments like linear probing, few-shot learning, and transfer learning using SSL features are missing. The paper would be much stronger if these standard experiments were included.
3. It might be good to compare NeSy + Prior SSL (like SwAV) and compare that to NeSy + GEDI in Section 6.4.

---

> ### Author Response · Authors · 2025-05-30
> **Answer to review**
>
> Thank you for the thorough, detailed assessment of our work and your valuable suggestions. We are pleased that you appreciated the content and the paper's contributions. We have submitted a revised version of our manuscript in which all modifications to the main text are highlighted in purple. Please find below detailed comments to the weaknesses you highlighted.
>
> **Scaling experiments** Thank you for raising this important point. As the reviewer correctly noted, we are limited in our ability to scale up the experiments due to the nature of EBM training. To the best of our knowledge, there are currently no well-established strategies for scaling EBMs beyond the datasets used in our paper. We would be very interested to hear the reviewer's thoughts or suggestions on potential directions for overcoming this limitation.
>
> **Linear probing** We have reported the supervised linear probing performance for all methods in Table 10 of Appendix N. Our approach is competitive with the best-performing baseline, Barlow Twins, which belongs to the family of feature decorrelation methods (referred to as negative-free SSL in our paper). Notably, our method outperforms the clustering-based competitor, SwAV.
>
> **Neuro-symbolic (NeSy) experiments** Thank you for this suggestion, which has helped improve the completeness of the NeSy experiments. We have added the additional baseline, NeSy + SwAV, to Table 6 in the main paper. This combination shows inferior performance compared to NeSy + GEDI. We attribute this to GEDI’s ability to identify concepts more robustly, aided by the regularizing effect of EBM training, which is not present in the clustering-based SwAV method.
>
> We hope we have addressed your concerns effectively. Please let us know if there is anything you would like to discuss further.

---

### Decision · Action_Editor_FnrF · 2025-07-16

**Recommendation:** Accept as is

**Audience:**

Yes

**Audience Explanation:**

Self-supervised learning techniques are of central importance for modern machine learning and thus of general interest. The authors have gone to great lengths to unify and explain the methodology. This is much appreciated.

**Claims And Evidence:**

Yes

**Claims Explanation:**

The reviewers have thoroughly read the experimental section and appreciate it. There is some concern about the size datasets investigated. However, for energy based models, the datasets can be considered large enough.